# Twin Evolution with Meta Preference Optimization for Semi-Supervised Learning of Large Language Models

## Abstract

Large Language Models (LLMs) have demonstrated remarkable capabilities across various domains, yet their adaptation to specific downstream tasks remains challenging due to limited labeled data. Although post-training methods (*e.g.*, SFT, DPO) have proven effective, they face significant limitations due to the scarcity of labeled data. In this paper, we present TwinEvol, a framework that treats downstream task training and evaluation as complementary, co-evolving submodules. TwinEvol introduces an evaluation agent that co-evolves with the main model; this agent is not a static external module but rather self-iterates and evolves through continuous interaction with the generation LLM after iterative calibration. The agent facilitates more nuanced assessment during downstream adaptation, incorporating hard negative mining and meta-preference optimization to achieve comprehensive feedback and effective knowledge transfer. Through an iterative twin evolution process, the framework establishes a self-reinforcing cycle that effectively propagates knowledge from labeled to unlabeled data while maintaining task alignment. Experiments across various downstream tasks demonstrate that TwinEvol achieves superior performance compared to existing methods. Our code is available at https://anonymous.4open.science/r/TwinEvol/.

## 1 Introduction

Large Language Models (LLMs) have demonstrated remarkable capabilities in general domains (Achiam et al., 2023; Wang et al., 2024b), yet their adaptation to specific downstream tasks remains a critical challenge (Xie et al., 2024; Luo et al., 2024; Lin et al., 2024), particularly in real-world scenarios where high-quality annotated data is scarce (Honovich et al., 2023; Kung et al., 2023; Cheng et al., 2024b). Current mainstream post-training optimization approaches, such as Supervised Fine-tuning (SFT) and preference optimization methods (Rafailov et al., 2024; Ethayarajh et al., 2024), are constrained by their dependence on high-quality data (Bhatt et al., 2024; Rafailov et al., 2024; Ethayarajh et al., 2024). In most real-world situations, we would face a mix of limited human-response data and abundant unannotated data (Zhu, 2005; Gu et al., 2025). To overcome these constraints, researchers have begun exploring semi-supervised approaches that leverage limited labeled data to guide learning from abundant unlabeled data (Luo et al., 2024).

Existing methods face several challenges: ❶ Traditional knowledge injection paradigms (such as SFT-driven pseudo-labeling) (Luo et al., 2024; Wang et al., 2025; Xia et al., 2024) struggle to fully extract and utilize complex negative feedback signals embedded in unlabeled data, limiting the model's ability to learn deeply from self-exploration. ❷ While preference learning methods introduce comparative mechanisms, they become susceptible to alignment drift when relying on external or static evaluators disconnected from the model's evolution (Son et al., 2024), potentially providing delayed, ineffective, or even erroneous feedback signals (Ramé et al., 2024). The core issue lies in the lack of organic integration between the model's **generation capability** and **evaluation capability**, which hinders efficient and reliable knowledge transfer from limited high-quality data to vast unlabeled datasets.

To address these challenges, we propose **TwinEvol**, a framework that enables the co-evolution of LLM generation and evaluation capabilities, designed to stimulate and calibrate the LLM's inherent

evaluation mechanism. Eliminating dependency on fixed external evaluators, TwinEvol empowers LLMs to develop dynamic, task-aligned preference recognition abilities.

Its innovations manifest in three dimensions: **Architecturally**, we design a dual-branch co-evolutionary architecture where the main LLM generates diverse exploratory responses with a co-evolving `Evaluative Agent`. This Evaluative Agent is not a static external module but a dynamic component that, after initial calibration with limited high-quality annotated data,

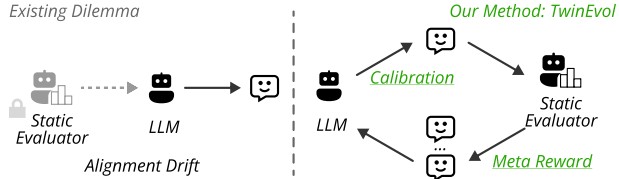

Figure 1: Conceptual overview of TwinEvol. (Left) Traditional methods rely on static evaluators, leading to outdated feedback and alignment drift. (Right) TwinEvol establishes a co-evolutionary loop to facilitate mutual improvement.

iteratively learns and evolves through continuous interaction with the generation module, forming an efficient knowledge transfer and self-improvement cycle. **Strategically**, TwinEvol establishes a composite negative feedback and multi-granular supervision system. Hard Negative Mining enables the LLM to extract subtle yet critical differences, enhancing its discrimination and improvement capabilities. Meanwhile, the Meta-Preference Optimization (MetaPO) algorithm achieves efficient supervision information aggregation, with theoretical analysis demonstrating its superior convergence properties and noise robustness. **Dynamically**, TwinEvol implements a Curriculum Evolution mechanism where the LLM's outputs train and validate the Evaluative Agent's judgment accuracy, while the Agent's precise feedback continuously optimizes the LLM's generation strategy. This iterative, interdependent update process forms a symbiotic co-evolution that enables synchronous enhancement of generation capabilities and internal evaluation standards, progressively achieving higher-level task alignment and performance improvement. Experimental validation across multiple downstream tasks demonstrates that TwinEvol significantly outperforms existing semi-supervised methods.

Our contributions can be summarized as follows: ❶ **Perspective:** We conceptualize LLM generation and evaluation capabilities as having a symbiotic relationship (Twin) and implement their iterative self-cyclical improvement. ❷ **Methodology:** We design TwinEvol, a co-evolutionary framework with MetaPO algorithm to achieve supervision information fusion and model adaptive evolution, with theoretical guarantees. ❸ **Performance:** We validate our method's effectiveness across multiple datasets, demonstrating significant performance improvements.

## 2 PRELIMINARIES

**Supervised Fine-Tuning** (SFT) represents the most straightforward approach for adapting LLMs to downstream tasks, aiming to align the model's outputs with desired responses. Formally, given a dataset $\mathcal{D} = \{X_i, Y_i\}_{i=1}^{N_l}$, where $N_l = \mathcal{N}(\mathcal{D})$ is the number of labeled samples in $\mathcal{D}$, $X_i$ is the input task and $Y_i$ is the corresponding expected response. SFT optimizes the model through a token-by-token loss minimization process. This procedure effectively injects knowledge. While traditional SFT focuses primarily on positive feedback,TwinEvol makes robustness improvement through adding comprehensive learning from negative LLM-generated pseudo samples.

**Semi-supervised Fine-Tuning** addresses scenarios with both labeled and unlabeled data. Given a labeled dataset $\mathcal{D}_{\text{labeled}} = \{(X_i, Y_i)\}_{i=1}^{N_l}$ and an unlabeled dataset $\mathcal{D}_{\text{unlabeled}} = \{X_i\}_{i=1}^{N_u}$, where $N_l \ll N_u$, the goal is to leverage limited labeled data as seeds to guide learning on unlabeled data. While existing approaches focus on direct knowledge transfer through self-training or consistency regularization, TwinEvol establishes a co-evolutionary mechanism between training and evaluation, creating a self-reinforcing *knowledge flywheel*, enabling more effective knowledge propagation.

**Preference Optimization** aims to align LLMs with human preferences through pairwise comparison learning. While DPO (Rafailov et al., 2024) has shown effectiveness in preference alignment using reference models, SimPO (Meng et al., 2024) proposes a more streamlined approach by removing reference model dependencies. The key difference between TwinEvol and existing preference optimization methods is that we use a co-evolutionary framework to optimize the preference learning, which is more effective than traditional preference optimization methods. GRPO (Shao et al., 2024) leverages group-wise comparisons but focuses on reinforcement learning to enhance long-

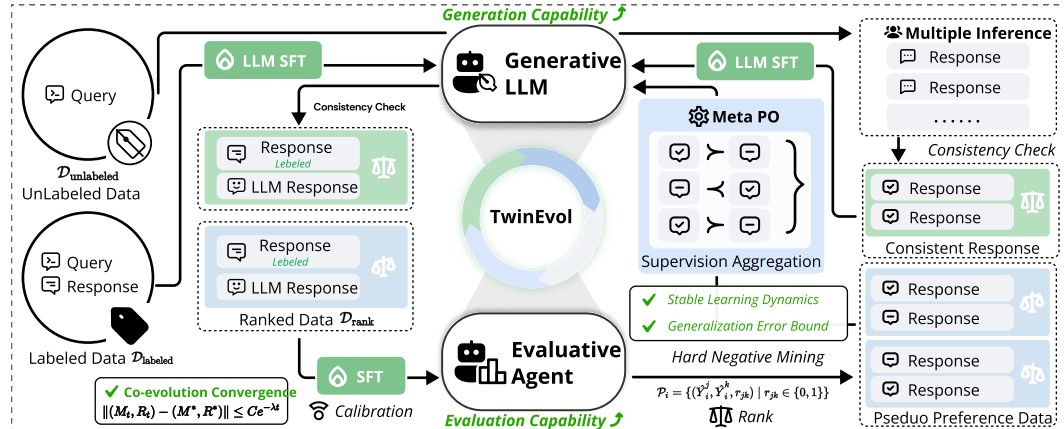

Figure 2: Overview of TwinEvol. The framework integrates a generative LLM with a co-evolved `Evaluative Agent` in a symbiotic relationship. The LLM processes both labeled and unlabeled data to generate responses, while the `Evaluative Agent` provides preference annotations through hard negative mining for MetaPO optimization. This establishes a twin evolution cycle where both components iteratively enhance each other's capabilities.

chain reasoning. Unlike GRPO which relies on rule-based verifiers and optimizes a group-relative objective, MetaPO is designed for semi-supervised adaptation, utilizing a co-evolved Evaluative Agent to provide pairwise preference signals from multiple generated responses.

## 3 METHODOLOGY

### 3.1 OVERVIEW

To handle hybrid data scenarios effectively, we propose TwinEvol. The core innovation of TwinEvol lies in organically integrating the model's generative and evaluative capabilities, forming a co-evolving system between the generative LLM and the `Evaluative Agent`. This framework creates an *iterative improvement cycle* that facilitates knowledge propagation from labeled to unlabeled data. As illustrated in Figure 2, our framework comprises two principal components: the generative LLM and the co-evolved `Evaluative Agent`. The framework's operation depends on the seamless coordination of several critical processes: the construction and training of the `Evaluative Agent` (Section 3.2), the learning of the generative LLM (Section 3.3), and our approach to Effective Meta Preference Optimization (Section 3.4), which employs hard negative mining and the MetaPO algorithm to refine model alignment. This co-evolutionary process culminates in a curriculum learning strategy (Section 3.5), enabling the system to improve continuously.

### 3.2 CO-EVOLVING EVALUATIVE AGENT

In semi-supervised scenarios, effectively leveraging limited labeled data to generate quality pseudo-preference signals for abundant unlabeled data is crucial. As shown in Figure 3, the `Evaluative Agent` serves as a dynamic quality assessor for model-generated responses. For a pair of responses $Y_0$ and $Y_1$, the `Evaluative Agent` determines their preference relationship, which can be either a tier relationship (*i.e.*, $Y_0 \sim Y_1$) or a strict preference (*i.e.*, $Y_0 \succ Y_1$).

The `Evaluative Agent` is trained through SFT using labeled data. Specifically, for each labeled data pair $(Q_i, Y_i)$, where $Q_i$ is the query and $Y_i$ is the annotated response, we generate model responses $\hat{Y}_i$ using the LLM. We then perform consistency checking by extracting and comparing answers. This process leads to either tier annotations (when responses are of similar quality) or preference annotations (when the labeled response is deemed superior). We identify cases where the model's generated response $\hat{Y}_i$ is inconsistent with (and typically inferior to) the

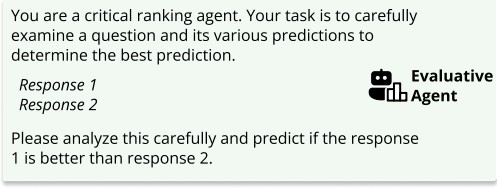

Figure 3: Instruction of the Evaluative Agent.

ground truth $Y_i$. We term these inconsistent preference annotations and use them to construct preference pairs $(Y_i \succ \hat{Y}_i)$ as training data for the Evaluative Agent:

$$\mathcal{D}_{\text{rank}} = \{(Q_i, Y_i, \hat{Y}_i, r_i \mid Y_i \succ \hat{Y}_i)\}. \tag{1}$$

The `Evaluative Agent` is then tuned by SFT on $\mathcal{D}_{\text{rank}}$. The agent is iteratively trained to learn preference relationships on downstream tasks, providing more accurate quality assessment for model-generated responses. Notably, after each update of the main LLM, new preference data $\mathcal{D}_{\text{rank}}$ is generated to train the `Evaluative Agent`, a process we refer to as **calibration**. This iterative calibration ensures that the `Evaluative Agent` continuously adapts to the evolving capabilities of the main model.

### 3.3 CO-EVOLVING GENERATIVE LLM

In the TwinEvol co-evolutionary framework, the main LLM could effectively utilize indirect feedback signals provided by the `Evaluative Agent` based on its generated responses. The training process of the LLM is as follows. As a warm-up, the LLM $\mathcal{M}$ first undergoes SFT using labeled data $\mathcal{D}_{\text{labeled}}$ to establish foundation domain adaptation capabilities.

In the adaptation process, we should make use of the unlabeled data $\mathcal{D}_{\text{unlabeled}}$ to further enhance the LLM's performance, which is conducted as follows. First, given an input task $X_i \in \mathcal{D}_{\text{unlabeled}}$, we generate multiple diverse responses through sampling from the $\mathcal{M}$:

$$\hat{\mathcal{Y}}_i = \{\hat{Y}_i^k = \mathcal{M}(X_i; \theta) \mid k = 1, \ldots, K\}, \tag{2}$$

where $K$ is the number of generated responses, which is set to 4 by default and examined in Section 4.3. Then, we classify the responses $\{\hat{\mathcal{Y}}_i\}_{i=1}^{N_u}$ into consistent set ($\mathcal{D}_{\text{con}}$) and inconsistent set ($\mathcal{D}_{\text{incon}}$) based on response consistency. The consistencies are derived from answer comparisons. For tasks with standard answers (e.g., multiple-choice), we use exact matching based on predefined rules; for open-ended generation, we employ LLM-based judgment to determine consistency. We provide detailed examples and a Case Study in Appendix.

The model then evolves through a two-pronged approach. For consistent responses ($\mathcal{D}_{\text{con}}$), we apply standard SFT training. For inconsistent responses ($\mathcal{D}_{\text{incon}}$), we leverage the `Evaluative Agent` to obtain pseudo-preferences and conduct MetaPO training (will be discussed in Section 3.4.2). The complete generative LLM update is as:

$$\mathcal{M}_{j+1} \leftarrow \text{MetaPO}(\mathcal{M}_j', \mathcal{D}_{\text{incon}}) \leftarrow \text{SFT}(\mathcal{M}_j, \mathcal{D}_{\text{con}}), \tag{3}$$

where $\mathcal{M}_j$ represents the model at iteration $j$, and $\mathcal{M}_j'$ is the model after SFT.

**Discussion.** This dual-branch architecture establishes the foundation for our co-evolution framework, where the `Evaluative Agent` provides quality assessment for LLM outputs, while the LLM generates diverse responses for `Evaluative Agent` training. This symbiotic relationship forms the basis for the twin evolution process detailed in the following section.

### 3.4 EFFECTIVE META PREFERENCE OPTIMIZATION

Traditional preference optimization approaches often rely on limited negative samples, which constrain the model's ability to learn from diverse error patterns. This limitation is particularly pronounced in semi-supervised learning scenarios. To address this challenge and enable robust optimization across the abundant unlabeled data, we introduce effective Meta Preference Optimization to provide richer alignment signals through diversified negative feedback. SimPO (Meng et al., 2024) can be viewed as a special case of MetaPO where $K = 2$ (a single pair). MetaPO generalizes this to $K > 2$, leveraging multiple generated responses to construct a richer set of negative pairs, which is crucial for stabilizing optimization in semi-supervised scenarios with noisy signals. Unlike simple rejection sampling that only utilizes positive samples, MetaPO leverages the relative preferences between multiple generated responses, providing richer supervision signals and enhanced stability.

### 3.4.1 HARD NEGATIVE MINING

Relying solely on single or randomly generated negative samples may not provide the model with sufficiently robust and targeted learning signals. Therefore, to fully leverage the learning potential from model outputs, we utilize multiple generations $\hat{\mathcal{Y}}_i$ from the LLM, and employ the `Evaluative Agent` to rank them for hard negative mining:

$$\mathcal{P}_i = \{(\hat{Y}_i^j, \hat{Y}_i^k, r_{jk}) \mid r_{jk} \in \{0,1\}, 1 \leq j \neq k \leq K\}, \tag{4}$$

where $r_{jk} = 1$ indicates $\hat{Y}_i^j$ is preferred over $\hat{Y}_i^k$, and this preference judgment is determined by the `Evaluative Agent`. Notably, as the `Evaluative Agent` and the model co-evolve through iterative training, the agent becomes increasingly capable of distinguishing subtle differences between various model-generated responses, resulting in more informative preference signals.

### 3.4.2 META PREFERENCE OPTIMIZATION

To effectively utilize the hard negative samples $\{\mathcal{P}_i\}_{i=1}^{N_u}$ and align the model with downstream tasks, we propose Meta Preference Optimization (MetaPO). Different from existing preference optimization methods (Rafailov et al., 2024; Ethayarajh et al., 2024), MetaPO achieves comprehensive consideration of multiple preference pairs, enabling more comprehensive preference optimization.

For MetaPO, we first introduce the length-normalized reward function (Meng et al., 2024):

$$r_\theta(X_i, \hat{Y}_i) = \frac{\beta}{|\hat{Y}_i|} \log \pi_\theta(\hat{Y}_i | X_i), \tag{5}$$

where $\beta$ is a constant controlling the reward scale (default to 1, examined in Section 4.3), and $|\hat{Y}_i|$ represents the response length. For a preference pair with winner response $\hat{Y}_i^w$ and loser response $\hat{Y}_i^l$, their reward difference should exceed a margin $\gamma$ (set to 1 following common practice): $r_\theta(X_i, \hat{Y}_i^w) - r_\theta(X_i, \hat{Y}_i^l) > \gamma$. The complete MetaPO objective is defined as:

$$\mathcal{L}_{\text{MetaPO}}(\pi_\theta) = -\frac{1}{N} \sum_{i=1}^N \sum_{(\hat{Y}_i^w, \hat{Y}_i^l) \in \mathcal{B}_i} \log \sigma(r_\theta(X_i, \hat{Y}_i^w) - r_\theta(X_i, \hat{Y}_i^l)). \tag{6}$$

Through MetaPO, we achieve comprehensive optimization across multiple types of negative feedback while maintaining stable model performance improvements. This approach enables robust optimization against diverse pseudo-responses, leading to more consistent model enhancement.

**Theoretical Analysis.** We provide a comprehensive theoretical foundation for TwinEvol, establishing its advantages through three complementary theoretical perspectives. ❶ We demonstrate that MetaPO achieves significant gradient variance reduction compared to single-pair optimization approaches, leading to more stable learning dynamics (Theorem 1). ❷ We establish a generalization error bound within the PAC-Bayes framework, quantifying how MetaPO's multi-pair mechanism tightens performance guarantees (Theorem 2). ❸ We analyze the convergence properties of our co-evolutionary framework, proving exponential convergence to optimal performance under well-defined conditions (Theorem 3). Together, these theoretical results establish a rigorous basis for understanding TwinEvol's empirical effectiveness in semi-supervised learning scenarios.

**Theorem 1 (Stable Learning Dynamics).** *Denote $\mathcal{L}_{SimPO}$ as the loss function of SimPO, which only generates one pair of responses per sample. Assuming these pairs $(\hat{Y}_i^w, \hat{Y}_i^l) \in \mathcal{B}_i$ are all independent, we have:*

$$\frac{Var[\nabla \mathcal{L}_{\text{MetaPO}}]}{Var[\nabla \mathcal{L}_{\text{SimPO}}]} = \mathcal{O}\left(\frac{1}{K^2}\right). \tag{7}$$

*Proof Sketch.* Let $M_i = |\mathcal{B}_i|$ be the number of pairs of contrastive pairs $(\hat{Y}_i^w, \hat{Y}_i^l)$ for each sample $X_i$. The gradient of MetaPO loss with respect to the parameters $\theta$ can be written as:

$$\nabla_\theta \mathcal{L}_{\text{MetaPO}} = -\frac{1}{N} \sum_{i=1}^N \left[ \frac{1}{M_i} \sum_{m=1}^{M_i} \sigma\left(r_\theta(X_i, \hat{Y}_m^l) - r_\theta(X_i, \hat{Y}_m^w)\right) \left(\nabla_\theta r_\theta(X_i, \hat{Y}_m^w) - \nabla_\theta r_\theta(X_i, \hat{Y}_m^l)\right) \right]. \tag{8}$$

For each sample $X_i$, the gradient can be viewed as the average of $M_i$ independent gradients, which makes the overall variance only $1/M_i$ of the variance of the SimPO loss gradient. Since $M = \mathcal{O}(K^2)$, it's straightforward to get to the conclusion of Theorem 1. The detailed proof is provided in Appendix B.1. □

This substantial variance reduction yields multiple advantages. Firstly, it fosters smoother and more stable learning dynamics. By framing the LLM's learning process as a trajectory on the $\theta$-parameterized statistical manifold. Secondly, a more consistent gradient direction enables the model to converge more directly and efficiently towards high-quality solution regions. This also reduces sensitivity to hyperparameter choices, such as the learning rate, rendering the training process more robust. Finally, by meticulously comparing multiple similar yet distinct candidate solutions and engaging in a form of *self-calibration*, MetaPO can more precisely capture and learn from subtle preference distinctions.

**Theorem 2** (**Generalization Error Bound**). *Let $\mathcal{R}(\pi_\theta)$ be the expected risk and $\hat{\mathcal{R}}(\pi_\theta)$ the empirical risk, then with probability $1 - \delta$, we have:*

$$\mathcal{R}(\pi_\theta) \leq \hat{\mathcal{R}}(\pi_\theta) + \mathcal{O}\left(\sqrt{\frac{\mathrm{KL}(\pi_\theta \| \pi_{ref}) + \ln(NK^2/\delta)}{NK^2}}\right) . \tag{9}$$

*where $\mathrm{KL}(\pi_\theta \| \pi_{ref})$ represents the KL-divergence between $\pi_\theta$ and base reference model $\pi_{ref}$.*

The proof of Theorem 2 can be found in Appendix B.2. This theorem provides theoretical guarantees for MetaPO's generalization performance. The bound shows the influence of the sample size $N$ and the number of generated responses $K$ on the generalization error. A larger $K$ yields up to $\binom{K}{2}$ preference pairs per input, effectively increasing the preference information used for training. This is reflected in the $NK^2$ term in the denominator of the bound, indicating that increasing $K$ generally tends to tighten the generalization bound. This complements the stable learning dynamics achieved with larger $K$, as stated in Theorem 1. We will discuss the effect of $K$ on the generalization error in Section 4.3. The $\ln(K^2)$ factor in the numerator, relative to $K^2$ in the denominator, suggests that the beneficial impact of $K$ on the error rate can be significant, assuming the KL divergence is appropriately bounded.

In this part, we analyze the convergence properties. To better quantify the performance improvements of LLM and the `Evaluative Agent`, we assume that after $t$ iterations, the performance of LLM is denoted as $M_t$, and the performance of the `Evaluative Agent` as $R_t$. Let $M^*$ and $R^*$ represent their optimal performance levels, respectively. Here, $\tilde{R}$ signifies the baseline performance of the `Evaluative Agent`. Specifically, $M_t$ improves when $R_t \geq \tilde{R}$, and declines when $R_t < \tilde{R}$. The following theorem demonstrates that, under specific assumptions, the co-evolutionary process achieves convergence with a rapid convergence rate, providing a theoretical foundation for the consistent performance gains in Section 4.

**Theorem 3** (**Co-evolution Convergence**). *Let $\alpha_M, \alpha_R$ be positive constants. Under the following assumptions:*

$$M_{t+1} - M_t = \alpha_M(M^* - M_t)(R_t - \tilde{R}), \quad R_{t+1} - R_t = \alpha_R(R^* - R_t)M_t, \tag{10}$$
$$0 < \tilde{R} \leq R_0 \leq R^*, \quad 0 < M_0 \leq M^*, \quad \alpha_R M^* \leq 1, \quad \alpha_M(R^* - \tilde{R}) \leq 1,$$

*then there exist constants $C$ and $\lambda > 0$ such that the following inequality holds:*

$$\|(M_t, R_t) - (M^*, R^*)\| \leq Ce^{-\lambda t} . \tag{11}$$

The proof of Theorem 3 can be found in Appendix B.3.

This theorem provides a theoretical foundation for the exponential convergence of our co-evolutionary framework. The results directly support TwinEvol's core design principle of symbiotic improvement, where the LLM benefits from increasingly accurate preference signals from the `Evaluative Agent` ($R_t - \tilde{R}$), while the Agent's capabilities are enhanced proportionally to the LLM's performance ($M_t$). This mathematical formulation validates our framework's ability to create a self-reinforcing knowledge flywheel through iterative co-evolution.

## 3.5 CURRICULUM LEARNING IN TWINEVOL

TwinEvol introduces a curriculum learning paradigm to address the critical challenge of data scarcity in downstream tasks. Central to this paradigm is the synergistic co-evolution of a generative LLM and an `Evaluative Agent`. This dual-entity system, enhanced by hard negative mining and

Table 1: **Performance comparison** across different models on various datasets. Red numbers show improvements of our method compared to SFT baseline. Best performance is highlighted in **bold**.

| Method | MMLU | MMLU Pro | ARC | FPB | USMLE | PubMedQA | ConvFinQA | Avg. |
|---|---|---|---|---|---|---|---|---|
| Vanilla | 66.4 | 47.1 | 81.1 | 81.7 | 70.2 | 73.5 | 51.1 | 67.3 |
| SFT | 67.9 | 49.8 | 81.8 | 96.2 | 70.8 | 75.0 | 81.3 | 74.7 |
| AdaptLLM | – | – | – | 49.7 | 31.5 | 27.6 | 30.9 | – |
| InstructPT | – | – | – | 76.1 | 47.4 | 44.5 | 55.2 | – |
| MemoryLLM | 56.4 | 31.8 | 56.3 | 57.7 | 37.8 | 55.5 | 37.2 | 47.5 |
| RAG (BM25) | 66.6 | 37.4 | 80.8 | 83.7 | 69.3 | 69.0 | 63.4 | 67.2 |
| RAG (FAISS) | 66.5 | 38.8 | 81.3 | 82.5 | 69.1 | 71.5 | 64.6 | 67.8 |
| Hermes-3 | 63.6 | 37.9 | 74.9 | 73.9 | 54.5 | 68.5 | 54.9 | 61.2 |
| Reflection-Llama | 65.5 | 37.5 | 82.2 | 80.8 | 67.4 | 71.5 | 40.8 | 63.7 |
| SemiEvol | 68.8 | 50.3 | 83.4 | 96.2 | 71.6 | 76.0 | 82.4 | 75.5 |
| **TwinEvol** *Iter 1* | 69.1+1.2 | 50.5+0.7 | 83.6+1.8 | 96.8+0.6 | 71.2+0.4 | 76.0+1.0 | 83.0+1.7 | 75.7+1.0 |
| **TwinEvol** *Iter 2* | 69.4+1.5 | 50.6+0.8 | 84.2+2.4 | 97.2+1.0 | 71.8+1.0 | 76.0+1.0 | 83.2+1.9 | 76.1+1.4 |
| **TwinEvol** *Iter 3* | 69.2+1.3 | **50.8**+1.0 | 84.3+2.5 | 97.2+1.0 | **72.2**+1.4 | 76.5+1.5 | 83.7+2.4 | 76.3+1.6 |
| **TwinEvol** *Iter 4* | **69.7**+1.8 | **50.8**+1.0 | **84.7**+2.9 | **97.5**+1.3 | 72.0+1.2 | **77.0**+2.0 | **84.1**+2.8 | **76.5**+1.8 |

Table 2: **Performance comparison** across different model architectures and sizes. Red numbers indicate improvements over the SFT baseline. Best performance is highlighted in **bold**.

| Method | MMLU | MMLU Pro | ARC | FPB | USMLE | PubMedQA | ConvFinQA | Avg. |
|---|---|---|---|---|---|---|---|---|
| *Llama3.2 3B* | | | | | | | | |
| Vanilla | 59.2 | 22.4 | 68.1 | 62.0 | 40.1 | 59.0 | 28.8 | 48.5 |
| SFT | 61.2 | 41.1 | 73.9 | 92.8 | 63.9 | 71.0 | 65.8 | 67.1 |
| Reflection | 60.1 | 30.2 | 72.4 | 62.0 | 42.9 | 58.5 | 23.2 | 49.9 |
| SemiEvol | 62.0 | 41.6 | 75.1 | 93.4 | 64.2 | 72.5 | 68.0 | 68.1 |
| **TwinEvol** *Iter 4* | **63.1** | **42.3** | **76.5** | **94.1** | **65.8** | **74.0** | **70.2** | **69.4** |
| *Llama3.1 8B* | | | | | | | | |
| Vanilla | 66.4 | 47.1 | 81.1 | 81.7 | 70.2 | 73.5 | 51.1 | 67.3 |
| SFT | 67.9 | 49.8 | 81.8 | 96.2 | 70.8 | 75.0 | 81.3 | 74.7 |
| Reflection | 65.5 | 37.5 | 82.2 | 80.8 | 67.4 | 71.5 | 40.8 | 63.7 |
| SemiEvol | 68.8 | 50.3 | 83.4 | 96.2 | 71.6 | 76.0 | 82.4 | 75.5 |
| **TwinEvol** *Iter 4* | **69.7** | **50.8** | **84.7** | **97.5** | **72.0** | **77.0** | **84.1** | **76.5** |
| *Gemma2 9B* | | | | | | | | |
| Vanilla | 72.1 | 43.0 | 87.1 | 73.7 | 58.7 | 64.5 | 45.9 | 63.6 |
| SFT | 73.7 | 50.0 | 87.6 | 95.2 | 66.6 | 74.5 | 79.8 | 75.3 |
| Reflection | 73.0 | 41.5 | 86.5 | 75.5 | 58.7 | 66.2 | 50.1 | 64.5 |
| SemiEvol | 74.5 | 50.8 | 88.0 | 96.0 | 67.3 | 75.3 | 81.5 | 76.2 |
| **TwinEvol** *Iter 4* | **75.4** | **51.9** | **88.6** | **96.5** | **68.4** | **76.8** | **83.7** | **77.3** |

Meta Preference Optimization, establishes a virtuous cycle: ❶ The LLM generates progressively sophisticated data, ❷ Upon which the `Evaluative Agent` refines its evaluative acuity, ❸ In turn providing more precise and challenging feedback. This dynamic interplay inherently cultivates an emergent curriculum, where the system autonomously adapts its learning trajectory to its evolving capabilities. Rather than relying on static datasets or predefined stages, TwinEvol demonstrates how a system can bootstrap its own advancement, transforming limited labeled data into a catalyst for sustained knowledge acquisition and performance gains. This culminates in a robust framework for continuous adaptation, paving the way towards more autonomous and resource-efficient learning models. Algorithm 1 and Appendix D.1 detail the complete training procedure.

## 4 EXPERIMENTS

### 4.1 EXPERIMENT SETUP

*Foundation Models.* To validate the broad applicability of our approach, we conducted experiments using foundation models with varying architectures and parameter scales. Our selection includes

Table 3: **Ablation study** demonstrating the effectiveness of key components in TwinEvol. ✓indicates the component is included in the corresponding variant. Rank: dynamic ranking via co-evolving Evaluative Agent; MPO: Meta Preference Optimization; CONS: consistency checking for response classification; SFT: Supervised Fine-Tuning.

| Variant | Rank | MPO | CONS | SFT | MMLU | MMLU-Pro | ARC |
|---------|------|-----|------|-----|------|----------|-----|
| **Full** | ✓ | ✓ | ✓ | ✓ | **69.7** | **50.8** | **84.7** |
| V1 | | | | ✓ | 66.7 | 47.9 | 82.0 |
| V2 | | | ✓ | ✓ | 68.9 | 50.2 | 83.3 |
| V3 | | ✓ | | ✓ | 67.5 | 50.1 | 82.2 |
| V4 | ✓ | ✓ | | ✓ | 68.2 | 50.2 | 83.2 |
| V5 | Static | ✓ | ✓ | ✓ | 67.1 | 49.8 | 82.5 |
| V6 | ✓ | SimPO | ✓ | ✓ | 69.0 | 50.5 | 82.4 |

Llama3.1-8B (Dubey et al., 2024), Gemma2-9B (Team et al., 2024), and Llama3.2-3B (Dubey et al., 2024), with diverse architectures and scales.

***Evaluation and Implementation Details*** We follow (Luo et al., 2024) for model semi-supervised fine-tuning. For all semi-supervised methods (including TwinEvol and SemiEvol), we utilize the complete unlabeled dataset $\mathcal{D}_{\text{unlabeled}}$ as described in Appendix D.2. Our evaluation suite encompasses both general-purpose and specialized domain benchmarks. General datasets include MMLU (Hendrycks et al., 2020), MMLU-Pro (Wang et al., 2024d), and ARC (Clark et al., 2018), while domain-specific evaluation includes FPB (Malo et al., 2014), USMLE (Jin et al., 2021), Pub-MedQA (Jin et al., 2019), and ConvFinQA (Chen et al., 2022), which target specialized knowledge in finance and healthcare domains. See Appendix D.2 for comprehensive implementation details. Code is available at https://anonymous.4open.science/r/TwinEvol.

***Baseline Methods.*** Our experimental evaluation encompasses several categories of baseline approaches: *(1) Fundamental Techniques*, including Vanilla inference and Supervised Fine-tuning (SFT) methods; *(2) Self-Evolution Approaches*, comprising self-reflection methods like Reflection-Llama (Li et al., 2024), and data-augmented training methods such as Hermes-3 (Teknium et al., 2024) and Tulu-3 (Lambert et al., 2024); *(3) Domain Adaptation Techniques*, including AdaptLLM (Cheng et al., 2024b) and InstructPT (Cheng et al., 2024a) for specialized domain adaptation; *(4) Inference Enhancement Methods*, such as MemoryLLM (Wang et al., 2024c) for retrieval-augmented generation. To ensure a fair comparison, we use the same foundation models for the main results (Table 1 and Table 2). To ensure fair comparison, all baseline methods utilize foundation models with comparable parameter counts.

## 4.2 MAIN RESULT

***Cross-method Comparison*** As shown in Table 1, TwinEvol achieves consistent performance improvements across all benchmarks, demonstrating its effectiveness in leveraging both labeled and unlabeled data. It is worth noting that TwinEvol significantly outperforms state-of-the-art semi-supervised baselines utilizing the same amount of labeled and unlabeled data, demonstrating the superiority of our co-evolutionary framework. We get the following observations: ❶ **Performance Enhancement through Supervision.** Table 1 shows that SFT yields improvements by leveraging labeled data, while SemiEvol approaches achieve further gains by effectively utilizing both labeled and unlabeled data. ❷ **Limitations of Post-training Methods.** Post-training techniques (specifically Self-Evolution methods and Domain Adaptation methods) show marginal improvements or degradation due to distribution misalignment and insufficient capability enhancement. ❸ **Constraints of Adaptive Fine-tuning.** Adaptive fine-tuning shows limited improvement due to lower-quality data sources compromising instruction-following capabilities. ❹ **Consistent Improvement.** TwinEvol demonstrates consistent improvements through effective unlabeled data utilization and multi-model collaborative learning.

***Cross-architecture Analysis.*** Table 2 demonstrates TwinEvol's consistent performance gains across diverse parameter scales and architectures, underscoring its inherent scalability. TwinEvol exhibits a unique capability for performance equilibration, which effectively addresses architectural predispositions through reciprocal knowledge transfer. This is exemplified by its ability to enhance Gemma's domain-specific performance (in FPB/PubMedQA) while preserving its general capabilities.

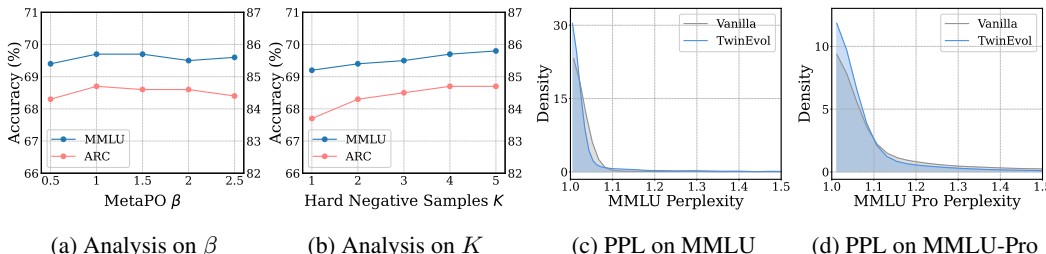

(a) Analysis on $\beta$     (b) Analysis on $K$     (c) PPL on MMLU     (d) PPL on MMLU-Pro

Figure 4: **Sensitivity analysis** of TwinEvol's performance, demonstrating its stability to hyperparameters, and **Perplexity analysis** on MMLU/MMLU-Pro.

## 4.3 ANALYSIS

***Ablation Study.*** We conducted ablation experiments to evaluate TwinEvol's components, with results shown in Table 3, which reveals: **(1) Full Model Performance.** The complete TwinEvol framework achieves optimal performance across all benchmarks, demonstrating effective component synergy. **(2) SFT Impact.** The performance gap between V1 (SFT-only) and baseline highlights supervised fine-tuning's crucial role, particularly evident in MMLU improvements. **(3) Consistency Selection.** V2's improvement over V1 demonstrates the effectiveness of consistency-based data filtering, with notable gains in MMLU-Pro and consistent improvements in other metrics. **(4) Preference Ranking.** The comparison between V4 and V3 shows the ranking model's contribution, yielding improvements. **(5) Co-evolution Necessity.** To validate the effectiveness of the dynamic Evaluative Agent, we design V5. Notably, V5 performs worse than V2. This reveals that naively adding preference optimization with a fixed evaluator can harm performance due to alignment drift. We further analyze the impact of evaluator accuracy in Appendix. Results show that TwinEvol remains robust even when the supervision signals contain noise (up to 30%), thanks to the MetaPO objective. Additionally, as shown in V6, MetaPO is better than SimPO in our tasks, corroborating our theoretical analysis.

***Sensitivity Analysis*** We conducted sensitivity analysis of the TwinEvol framework on MMLU and ARC datasets, focusing on the MetaPO coefficient $\beta$ and the number of hard negative samples $K$. As shown in Figure 4a and 4b, the model demonstrates robust performance across different values of $\beta$ and $K$. Performance improves with increasing $K$ before stabilizing at $K = 4$. Based on these observations, we set $\beta = 1$ and $K = 4$ as default configurations. The analysis reveals that TwinEvol exhibits low sensitivity to hyperparameter variations, demonstrating its stability.

***Perplexity Analysis.*** Figure 4c and 4d illustrate the perplexity distribution of TwinEvol compared to vanilla models. Our approach substantially reduces perplexity on both MMLU and MMLU-Pro, indicating enhanced model calibration and decision confidence. The consistent reduction across diverse benchmarks suggests effective knowledge integration rather than task-specific overfitting.

***Statability Analysis.*** Figure 5 presents our analysis of model inference stability. We employed GPT-4o to rephrase test instructions and conducted 5 independent evaluations, reporting both mean performance and standard deviation. The results demonstrate that TwinEvol maintains comparable stability to the original model.Details on the stability analysis are provided in Appendix C.2.

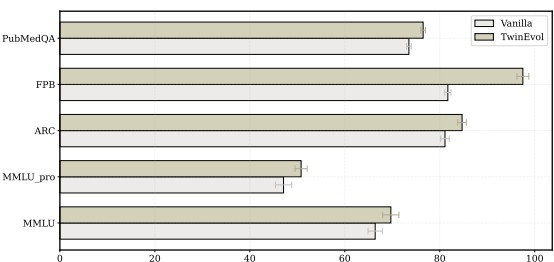

Figure 5: **Stability analysis** of TwinEvol.

***Computational Efficiency*** Despite introducing a dual-branch architecture, TwinEvol maintains competitive computational efficiency while delivering consistent performance improvements. Detailed computational efficiency analysis is provided in Appendix C.1.

## 5 RELATED WORK

***LLM Post-training*** is crucial for unlocking their domain adaptation capabilities and task generalization potential (Wang et al., 2024e; Jeong et al., 2024; Wang et al., 2024a). This stage effectively enhances model performance across multiple dimensions, including long-context reasoning (Zelikman et al., 2022), human alignment (Kaufmann et al., 2023; Rafailov et al., 2024), instruction following (Zhang et al., 2023), and domain-specific adaptation (Cheng et al., 2024b). Through post-training, LLMs can be effectively deployed in specialized domains such as healthcare (Jin et al., 2019), finance (Chen et al., 2022), and legal applications (Izzidien et al., 2024), making it a critical step in realizing their practical value. However, the scarcity of domain-specific data presents a significant challenge in real-world applications, leading to data efficiency concerns in LLM post-training (Tan et al., 2024; Xu et al., 2024b; Kundu et al., 2024).

***Data-efficient LLM Post-training*** aims to address real-world data scarcity, including data selection (Tsai et al., 2024; Zhou et al., 2024; Kim & Baek, 2024; Lu et al., 2024), data synthesis (Xu et al., 2024c;a; Dai et al., 2025), and model self-evolution (Madaan et al., 2024; Chen et al., 2024; You et al., 2024; Dong et al., 2024). While these methods have been extended to semi-supervised learning scenarios (Luo et al., 2024) with limited labeled and abundant unlabeled data. Existing semi-supervised post-training methods face critical challenges: the accumulation of pseudo-labeling errors, insufficient negative feedback mechanisms, and static paradigms lack continuous evolution capabilities. We distinguish TwinEvol from recent self-alignment works like Self-Rewarding LMs (Yuan et al., 2024), Meta-Rewarding LMs (Wu et al., 2025), and Semi-supervised Reward Modeling (He et al., 2024). While these methods focus on general alignment or reward modeling, TwinEvol targets semi-supervised downstream adaptation, leveraging specific unlabeled datasets through a robust co-evolutionary framework. In this work, TwinEvol leverages collaborative learning with MetaPO and curriculum learning for a self-reinforcing cycle of knowledge transfer.

## 6 CONCLUSION

In this paper, we present TwinEvol, a novel semi-supervised co-evolutionary framework for LLM adaptation that addresses the challenge of limited high-quality annotated data. Through its unique dual-branch architecture incorporating a generative LLM and a co-evolving Evaluative Agent, along with Hard Negative Mining and MetaPO optimization algorithm, TwinEvol effectively facilitates knowledge transfer and capability amplification from limited labeled data to abundant unlabeled data. Both theoretical analysis and extensive experimental results validate that TwinEvol significantly enhances model performance, offering a promising direction for LLM adaptation. Although TwinEvol introduces training-time complexity, it maintains standard inference efficiency and automates supervision mining, offering a favorable trade-off for data-scarce scenarios.

### REPRODUCIBILITY STATEMENT

For reproducibility purposes, we have made our code available at https://anonymous.4open.science/r/TwinEvol/. Also, we provided the detailed implementation details in Section 4.1, Appendix D.2 and Appendix C.

### ETHICS STATEMENT

Our research adheres to the ICLR Code of Ethics.The code and related materials will be appropriately released to ensure transparency and reproducibility of our work. All datasets used in this study are publicly available.

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

# Appendix

In this Appendix, we present comprehensive supplementary materials that underpin the results discussed in the main text. Section A provides a detailed reproducibility statement, Section B contains rigorous proofs of all theorems, Section C offers additional analyses on computational efficiency, stability, and further discussion, and Section D delivers full algorithmic and implementation details.

## Appendix Contents

## A    REPRODUCIBILITY

To increase reproducibility, we have provided all the details of TwinEvol in the Appendix. Our code is available at https://anonymous.4open.science/r/TwinEvol/ anonymously. We also commit to making our code and data publicly available.

## B    PROOF OF THEOREMS

### B.1    PROOF OF THEOREM 1

**Theorem 4.** *Denote $\mathcal{L}_{SimPO}$ as the loss function of SimPO, which only generates one pair of responses per sample. Assume that each $Y_i^k$ is generated independently, we have*

$$\frac{Var[\nabla\mathcal{L}_{\text{MetaPO}}]}{Var[\nabla\mathcal{L}_{\text{SimPO}}]} = \mathcal{O}(\frac{1}{K^2}) \tag{12}$$

*Proof.* Let's analyze this step by step:

1) First, let's consider SimPO which only uses a single preference pair per sample. For each input task $X_i$, let $\mathcal{B}_i = \{(\hat{Y}_m^w, \hat{Y}_m^l)\}_{m=1}^{M_i}$ be $M_i$ preference pairs, where $M_i = \mathcal{O}(K^2)$.

2) For SimPO with only a single pair $(\hat{Y}^w, \hat{Y}^l)$ per sample $X_i$, the gradient is:

$$\nabla_\theta\mathcal{L}_{\text{SimPO}} = -\frac{1}{N}\sum_{i=1}^{N}\left[\sigma(r_\theta(X_i,\hat{Y}^l) - r_\theta(X_i,\hat{Y}^w))\ (\nabla_\theta r_\theta(X_i,\hat{Y}^w) - \nabla_\theta r_\theta(X_i,\hat{Y}^l))\right] \tag{13}$$

3) To simplify notation, let's denote:

$$g_i(\hat{Y}^w, \hat{Y}^l) = \sigma(r_\theta(X_i, \hat{Y}^l) - r_\theta(X_i, \hat{Y}^w))(\nabla_\theta r_\theta(X_i, \hat{Y}^w) - \nabla_\theta r_\theta(X_i, \hat{Y}^l)) \quad (14)$$

and its variance $\sigma_g^2 = Var(g_i(\hat{Y}^w, \hat{Y}^l))$.

4) For SimPO, the gradient variance can be calculated as:

$$Var(\nabla_\theta \mathcal{L}_{\text{SimPO}}) = \frac{1}{N^2} \sum_{i=1}^{N} \sigma_g^2 = \frac{\sigma_g^2}{N} \quad (15)$$

5) For MetaPO, which utilizes multiple preference pairs, the gradient is:

$$\nabla_\theta \mathcal{L}_{\text{MetaPO}} = -\frac{1}{N} \sum_{i=1}^{N} \left[ \frac{1}{M_i} \sum_{m=1}^{M_i} \sigma(r_\theta(X_i, \hat{Y}_m^l) - r_\theta(X_i, \hat{Y}_m^w)) \ (\nabla_\theta r_\theta(X_i, \hat{Y}_m^w) - \nabla_\theta r_\theta(X_i, \hat{Y}_m^l)) \right]$$

$$= -\frac{1}{N} \sum_{i=1}^{N} \left[ \frac{1}{M_i} \sum_{m=1}^{M_i} g_i(\hat{Y}_m^l, \hat{Y}_m^w) \right] \quad (16)$$

6) Due to the independence assumption of generated responses, the gradient variance of MetaPO is:

$$Var(\nabla_\theta \mathcal{L}_{\text{MetaPO}}) = \frac{1}{N^2} \sum_{i=1}^{N} \frac{1}{M_i^2} \sum_{m=1}^{M} \sigma_g^2 = \mathcal{O}\left( \frac{\sigma_g^2}{NK^2} \right) \quad (17)$$

7) Therefore, the ratio of variations is:

$$\frac{Var[\nabla \mathcal{L}_{\text{MetaPO}}]}{Var[\nabla \mathcal{L}_{\text{SimPO}}]} = \mathcal{O}\left( \frac{1}{K^2} \right) \quad (18)$$

This result demonstrates that MetaPO significantly reduces gradient variance compared to SimPO by a factor of $\mathcal{O}(\frac{1}{K^2})$. This reduction in variance leads to more stable training and potentially faster convergence. The quadratic relationship with K suggests that even a modest increase in the number of generated responses can substantially improve training stability. $\square$

## B.2 PROOF OF THEOREM 2

**Theorem 5.** *Let $\mathcal{R}(\pi_\theta)$ be the expected risk and $\hat{\mathcal{R}}(\pi_\theta)$ the empirical risk, then with probability $1 - \delta$, we have*

$$\mathcal{R}(\pi_\theta) \leq \hat{\mathcal{R}}(\pi_\theta) + \mathcal{O}\left( \sqrt{ \frac{KL(\pi_\theta \| \pi_{ref}) + \ln(\frac{NK^2}{\delta})}{NK^2} } \right) \quad (19)$$

*where $KL(\pi_\theta \| \pi_{ref})$ represents the KL-divergence between $\pi_\theta$ and base reference model $\pi_{ref}$.*

*Proof.* First, we define the dataset $\tilde{\mathcal{D}} = \{(X_i, \hat{Y}_m^w, \hat{Y}_m^l)\}_{1 \leq i \leq n, 1 \leq m \leq M_i}$ and the loss function $\ell(\theta; X_i, \hat{Y}_i^w, \hat{Y}_m^l) = -\log \sigma \left( r_\theta(X_i, \hat{Y}_m^w) - r_\theta(X_i, \hat{Y}_m^l) \right)$. Let $\mathcal{D}$ be the distribution of the input $X$. Then we can calculate the empirical risk $\hat{\mathcal{R}}(\pi_\theta) = \frac{1}{N} \sum_{i=1}^{N} \frac{1}{M_i} \sum_{m=1}^{M_i} \ell(\theta; X_i, \hat{Y}_i^w, \hat{Y}_m^l)$ and the expected risk $\mathcal{R}(\pi_\theta) = \mathbb{E}_{X \sim \mathcal{D}, \hat{Y}^w, \hat{Y}^l \sim \pi_\theta}[\ell(\theta; X, \hat{Y}^w, \hat{Y}^l)]$.

Under the PAC-Bayes Framework, using McAllester's bound (McAllester, 2003), with probability $1 - \delta$, we have

$$\mathcal{R}(\pi_\theta) \leq \hat{\mathcal{R}}(\pi_\theta) + \sqrt{ \frac{KL(\pi_\theta \| \pi_{\text{ref}}) + \log \frac{\tilde{N}}{\delta}}{2(\tilde{N} - 1)} } \quad (20)$$

where $\tilde{N} = \sum_{i=1}^{N} M_i$ is the total amount of data $(X_i, \hat{Y}_m^w, \hat{Y}_m^l)$ we used and $\pi_{\text{ref}}$ is the reference policy.

Since $M_i = \mathcal{O}(K^2)$, we can get that with probability $1 - \delta$, the following inequality holds.

$$\mathcal{R}(\pi_\theta) \leq \hat{\mathcal{R}}(\pi_\theta) + \mathcal{O}\left(\sqrt{\frac{\mathrm{KL}(\pi_\theta \| \pi_{\mathrm{ref}}) + \ln(\frac{NK^2}{\delta})}{NK^2}}\right). \tag{21}$$

$\square$

**Theoretical Implications.** This generalization bound provides several key insights into MetaPO's behavior. First, the error decreases at rate $O(1/\sqrt{N})$, which achieves the optimal convergence rate in statistical learning theory. This indicates MetaPO's strong statistical efficiency in utilizing training samples.

**Trade-offs in Response Generation.** The $O(\sqrt{K \log K / N})$ term reveals an important trade-off in the number of generated responses $K$. While Theorem 1 shows that larger $K$ reduces gradient variance quadratically, the generalization bound suggests a mild increase in error with $K$. The logarithmic dependence on $K$ indicates this trade-off favors moderate values that balance variance reduction and generalization.

**Model Complexity Considerations.** The bound's implicit dependence on model dimension $d$ through the covering number illustrates the relationship between model complexity and the theoretical guarantees. This provides a formal characterization of how model architecture choices affect the learning process.

### B.3 PROOF OF THEOREM 3

**Theorem 6** (Co-evolution Convergence). *Let $\alpha_M, \alpha_R$ be positive constants. Under the following assumptions:*

$$M_{t+1} - M_t = \alpha_M (M^* - M_t)(R_t - \tilde{R})$$
$$R_{t+1} - R_t = \alpha_R (R^* - R_t) M_t$$
$$0 < \tilde{R} \leq R_0 \leq R^* , \; 0 < M_0 \leq M^* \tag{22}$$
$$\alpha_R M^* \leq 1 , \; \alpha_M (R^* - \tilde{R}) \leq 1$$

*then there exist constants $C$ and $\lambda > 0$ such that the following inequality holds:*

$$\|(M_t, R_t) - (M^*, R^*)\| \leq C e^{-\lambda t}. \tag{23}$$

*Proof.* The given system is a discrete-time dynamical system, and its fixed point is $(M^*, R^*)$.

From the assumptions, we have

$$M^* - M_{t+1} = (M^* - M_t)(1 - \alpha_M(R_t - \tilde{R}))$$
$$R^* - R_{t+1} = (R^* - R_t)(1 - \alpha_R M_t) \tag{24}$$

Denote $A_t = 1 - \alpha_M(R_t - \tilde{R})$, $B_t = 1 - \alpha_R M_t$ for $t \in \mathbb{N}$. Then we have $0 \leq A_t, B_t \leq 1$, and for any $t > s > 0$,

$$M^* - M_{t+1} = (M^* - M_t) \cdot A_t = \cdots = (M^* - M_s) \cdot A_t A_{t-1} \cdots A_s$$
$$R^* - R_{t+1} = (R^* - R_t) \cdot B_t = \cdots = (R^* - R_s) \cdot B_t B_{t-1} \cdots B_s \tag{25}$$

Then, we can obtain that $\tilde{R} \leq R_t \leq R^*, 0 < M_t \leq M^*, A_t < \mu, B_t < \mu (t \geq t_0)$ for some $0 < \mu < 1$ and $t_0$, which is not hard to prove by induction from Equation 25. Therefore, we have

$$M^* - M_t \leq (M^* - M_{t_0})\mu^{t-t_0} , \quad R^* - R_t \leq (R^* - R_{t_0})\mu^{t-t_0} \tag{26}$$

which implies

$$\begin{aligned} \|(M_t, R_t) - (M^*, R^*)\| &= \sqrt{(M_t - M^*)^2 + (R_t - R^*)^2} \\ &\leq \sqrt{(M_{t_0} - M^*)^2 + (R_{t_0} - R^*)^2} \cdot \mu^{t-t_0} \\ &= C\mu^t = C' e^{-\lambda t} \end{aligned} \tag{27}$$

for some constant $C, C'$ and $\lambda > 0$. $\square$

## C  ADDITIONAL ANALYSIS

### C.1  COMPUTATIONAL EFFICIENCY ANALYSIS

To assess computational overhead and scaling, we conducted supplementary experiments across two model sizes (Llama3.2-3B and Llama3.1-8B), comparing per-iteration training time against standard SFT and a memory-augmented baseline.

Table 4: Computational resource analysis for SFT, MemoryLLM, and TwinEvol across model sizes.

| Method | Llama3.2-3B | Llama3.1-8B |
|---|---|---|
| SFT | 0.62 | 1.50 |
| MemoryLLM | 1.39 | 3.02 |
| TwinEvol | 0.80 | 1.98 |

As shown in Table 4, the per-iteration overhead of TwinEvol relative to SFT is approximately 29% on 3B and 32% on 8B. The overhead does not increase substantially with parameter count and remains well below MemoryLLM, indicating practical efficiency at larger scales. The reported time includes the overhead of the Evaluative Agent. Compared to SemiEvol, TwinEvol incurs a marginal increase in computation due to the dual-branch updates, but achieves significantly better performance. The initial SFT phase is a one-time cost shared by all fine-tuning methods.

Three design choices contribute to this efficiency. First, shared parameter initialization: both the LLM and the Evaluative Agent reuse the same foundation architecture, enabling efficient initialization and optimization. Second, selective preference optimization: MetaPO is applied only to inconsistent preference pairs, avoiding computation on already well-aligned data. Third, iterative co-evolution: as both models improve, later iterations learn more efficiently, partially amortizing the initial overhead.

Overall, the overhead is comparable to common alignment methods such as SFT/DPO, while TwinEvol delivers consistent gains in semi-supervised settings. The favorable scaling profile makes TwinEvol viable for real-world deployment at scale.

### C.2  STABILITY ANALYSIS

Table 5: Stability Analysis Results (Mean $\pm$ Std).

| Model | MMLU | MMLU_pro | ARC | FPB | PubMedQA |
|---|---|---|---|---|---|
| Vanilla | $66.4 \pm 1.50$ | $47.1 \pm 1.70$ | $81.1 \pm 0.90$ | $81.7 \pm 0.65$ | $73.5 \pm 0.50$ |
| TwinEvol | $\mathbf{69.7 \pm 1.70}$ | $\mathbf{50.8 \pm 1.27}$ | $\mathbf{84.7 \pm 0.90}$ | $\mathbf{97.5 \pm 1.25}$ | $\mathbf{76.5 \pm 0.50}$ |

To thoroughly evaluate the stability of TwinEvol, we conducted a comprehensive analysis comparing it with the vanilla model across five standard benchmarks. Table 5 presents detailed results showing both models' performance with standard deviations across 5 independent runs. The data reveals that TwinEvol maintains comparable or better stability than the vanilla model while achieving higher accuracy across all benchmarks.

The analysis demonstrates that TwinEvol's improvements are statistically significant and robust. The standard deviations remain comparable to or lower than the baseline model, indicating that our approach enhances performance without sacrificing stability. This is particularly noteworthy in specialized domains like FPB and PubMedQA, where TwinEvol shows substantial improvements while maintaining low variance. The consistent performance across multiple runs suggests that the twin evolution mechanism effectively stabilizes the learning process, likely due to the mutual reinforcement between the LLM and Evaluative Agent components.

## C.3 ROBUSTNESS UNDER NOISY UNLABELED DATA

We evaluate robustness to systematic noise in unlabeled data on ConvFinQA. To simulate semantic noise, we obfuscate and paraphrase a fraction (10%, 20%, 30%) of unlabeled queries using GPT-4o, and train all models on these noisy unlabeled sets while keeping the evaluation on the original test set. This setup tests stability under distribution corruption without changing the gold evaluation protocol.

Table 6: Robustness to semantic noise on ConvFinQA (Accuracy, %). Unlabeled queries are paraphrase-obfuscated at the indicated rates; evaluation is on the original test set.

| Method | 0% | 10% | 20% | 30% |
|---|---|---|---|---|
| SFT | 81.3 | - | - | - |
| SFT+SelfLabel | 81.9 | 80.9 | 79.5 | 77.8 |
| SemiEvol | 82.4 | 82.0 | 81.3 | 79.6 |
| TwinEvol | **84.1** | **83.6** | **83.1** | **82.9** |

The baseline methods degrade notably as noise increases, reflecting error accumulation in self-labeling. In contrast, TwinEvol exhibits graceful degradation, attributed to the `Evaluative Agent` and Meta Preference Optimization jointly suppressing error propagation. Under 30% noise, TwinEvol still surpasses the SFT model trained only on clean data, indicating robust extraction of signal from noisy unlabeled corpora.

## C.4 FURTHER DISCUSSION

***Overfitting.*** A potential concern is whether training the Evaluative Agent using only labeled data might lead to overfitting, given the typically small size of such datasets. However, the Evaluative Agent in our TwinEvol framework is designed to mitigate this risk. Firstly, each labeled example generates multiple preference pairs through comparisons with model-generated responses, which effectively expands the training set for the Evaluative Agent. Secondly, the iterative co-evolution process of TwinEvol continuously exposes the Evaluative Agent to new and evolving response patterns from the LLM. This dynamic training environment helps the Evaluative Agent to generalize better. Our empirical results, which demonstrate consistent performance improvements across diverse benchmarks, validate this approach and indicate effective generalization rather than overfitting of the Evaluative Agent.

***Comparison with Traditional SFT+PO Pipelines.*** It is crucial to highlight that traditional SFT+PO pipelines operate under significantly different data paradigms. These methods typically necessitate substantial volumes of human-annotated preference pairs for effective alignment. In contrast, TwinEvol is specifically engineered for semi-supervised learning scenarios characterized by a scarcity of labeled data and, importantly, the absence of human-generated preference feedback. This fundamental divergence in data availability and assumptions makes a direct, equitable comparison challenging. TwinEvol's primary contribution lies in its efficacy within these low-resource, semi-supervised contexts, where access to extensive preference datasets is not feasible.

***Robustness to Noisy Hard Negatives.*** In scenarios involving low-quality or noisy initial data, the risk of unreliable negative samples impacting preference optimization is a valid concern. TwinEvol incorporates several mechanisms to enhance robustness against such challenges. Firstly, an initial SFT phase on available labeled data establishes a foundational level of response quality. Secondly, the `Evaluative Agent`, trained on clean labeled examples, provides a more reliable source for preference judgments, guiding the LLM even when its own initial generations might be noisy. Thirdly, MetaPO's design, which aggregates signals from multiple preference pairs, inherently offers a degree of resilience against individual noisy or misleading negative samples. Finally, the iterative co-evolutionary process itself fosters a curriculum effect: as both the LLM and the `Evaluative Agent` improve, the quality of generated responses progressively refines, creating a virtuous cycle that mitigates the early-stage impact of noise. While no system is entirely immune to noise, these integrated strategies collectively contribute to TwinEvol's ability to navigate and improve even in imperfect data environments.

***Mitigation of Pseudo-Label Error Propagation.*** While pseudo-labeling inherently bears the risk of propagating erroneous signals, TwinEvol employs a multi-faceted defense strategy. First, the `Evaluative Agent` is initially trained on clean, human-annotated data (Eq. 1), establishing a robust foundation before generating pseudo-preferences. Second, the iterative co-evolution of the LLM and `Evaluative Agent` creates a virtuous cycle: as the LLM's outputs improve in quality, the `Evaluative Agent` receives more reliable training signals, progressively curbing error amplification. Third, MetaPO's aggregation over multiple diverse response pairs dilutes the impact of any single incorrect annotation, as formalized in Theorem 1. Finally, the emergent curriculum effect from this co-evolutionary process ensures that both models naturally transition from simpler to more complex data, further containing noise. Empirical results across benchmarks consistently validate the efficacy of these safeguards, demonstrating stable performance even under ambiguous or noisy conditions.

---

**Algorithm 1** Algorithm of TwinEvol

---

**Require**: Labeled dataset $\mathcal{D}_{\text{labeled}}$, Unlabeled dataset $\mathcal{D}_{\text{unlabeled}}$, LLM $\mathcal{M}$, `Evaluative Agent` (denoted as $\mathcal{A}$), Number of responses $K$, Number of iterations $T$;
**Ensure**: Fine-tuned LLM $\mathcal{M}'$

1: Initialize $\mathcal{M}$ and $\mathcal{A}$ with pre-trained weights
2: $\mathcal{M} \leftarrow \text{SFT}(\mathcal{M}, \mathcal{D}_{\text{labeled}})$
3: **for** iteration $t = 1$ to $T$ **do**
4:     *// `Evaluative Agent` Training Step*
5:     Initialize $\mathcal{D}_{\text{rank}} = \emptyset$
6:     **for** each labeled sample $(X_i, Y_i)$ in $\mathcal{D}_{\text{labeled}}$ **do**
7:         Generate response $\hat{Y}_i = \mathcal{M}(X_i)$
8:         **if** $Y_i \succ \hat{Y}_i$ **then**
9:             Add $(X_i, Y_i, \hat{Y}_i)$ to $\mathcal{D}_{\text{rank}}$
10:         **end if**
11:     **end for**
12:     Update `Evaluative Agent`: $\mathcal{A} \leftarrow \text{SFT}(\mathcal{A}, \mathcal{D}_{\text{rank}})$
13:     *// LLM Training Step*
14:     Initialize $\mathcal{D}_{\text{con}} = \emptyset$
15:     Initialize $\mathcal{D}_{\text{pref\_pairs}} = \emptyset$
16:     **for** each sample $X_i$ in $\mathcal{D}_{\text{unlabeled}}$ **do**
17:         Generate $K$ diverse responses: $\{\hat{Y}_i^k\}_{k=1}^K = \mathcal{M}(X_i)$
18:         Identify consistent responses $\hat{Y}_{i,\text{cons}}$ from $\{\hat{Y}_i^k\}$
19:         **if** such $\hat{Y}_{i,\text{cons}}$ exists for $X_i$ **then**
20:             Add $(X_i, \hat{Y}_{i,\text{cons}})$ to $\mathcal{D}_{\text{con}}$
21:         **end if**
22:         Construct preference pairs $\mathcal{P}_i = \{(\hat{Y}_i^w, \hat{Y}_i^l)\}$ from all $K$ responses $\{\hat{Y}_i^k\}$ using the updated `Evaluative Agent` $\mathcal{A}$
23:         Add all pairs from $\mathcal{P}_i$ to $\mathcal{D}_{\text{pref\_pairs}}$
24:     **end for**
25:     Update LLM via SFT: $\mathcal{M} \leftarrow \text{SFT}(\mathcal{M}, \mathcal{D}_{\text{con}})$
26:     Update LLM via MetaPO: $\mathcal{M} \leftarrow \text{MetaPO}(\mathcal{M}, \mathcal{D}_{\text{pref\_pairs}})$
27: **end for**
28: **return** $\mathcal{M}' = \mathcal{M}$

---

# D ADDITIONAL DETAILS

## D.1 ALGORITHM DETAILS

Algorithm 1 presents the complete training procedure of TwinEvol. The algorithm consists of three main components that operate in an iterative manner. First, Hard Negative Mining (lines 4-7) generates $K$ diverse responses for each input and constructs preference pairs, providing comprehensive supervision signals. Second, the `Evaluative Agent` Training Step (lines 8-12) leverages la-

beled data to create ranking annotations, enabling the `Evaluative Agent` to learn preference relationships. Finally, the LLM Training Step (lines 13-17) updates the model through both SFT on consistent responses and MetaPO on inconsistent ones, forming a twin evolution cycle. This iterative process naturally implements curriculum learning as both models progressively improve, where enhanced model capabilities lead to better quality training signals, which in turn enable further model improvements in subsequent iterations.

## D.2 Implementation Details

We provide comprehensive implementation details to ensure reproducibility of our experiments. All experiments were conducted on NVIDIA Hopper GPUs with 80GB memory. The implementation leverages PyTorch with CUDA and mixed precision training (FP16/BF16).

For the foundation models, we employed different sequence length configurations to accommodate model architectures. Llama3.1-8B and Llama3.2-3B operate with a maximum sequence length of 4096 tokens, while Gemma2-9B extends to 8192 tokens to leverage its enhanced context handling capabilities.

The training process utilizes a carefully tuned set of hyperparameters optimized for model performance. We employ a learning rate of 2e-5 with a cosine decay schedule, complemented by 100 warmup steps. The optimization process incorporates weight decay of 0.01 and gradient clipping at 1.0. To balance computational efficiency and training stability, we set gradient accumulation steps to 4 and run the training for 3 epochs. For MetaPO optimization, we set $\beta = 1.0$ and $K = 4$ based on our sensitivity analysis (Section 4.3). The `Evaluative Agent` model follows identical hyperparameters except for an increased learning rate of 5e-5 to facilitate faster adaptation. In the decoding process, we use temperature sampling with temperature 1.

Our data processing pipeline implements several key optimizations. Input sequences are tokenized using model-specific tokenizers with the aforementioned maximum sequence lengths. The clean set uses an entropy-based filtering with a ratio of $50\%$ following previous work.

For consistent and fair comparison, our data partitioning follows the established protocol from SemiEvol (Luo et al., 2024), with a ratio of 2:6:2 for labeled data, unlabeled data, and test data, respectively.

**Consistency Check Procedure.** We generate multiple candidate responses for each unlabeled data point and apply consistency validation. In our main experiments (e.g., MMLU, ARC), we primarily utilize rule-based extraction which is efficient and effective for these structured tasks. However, the TwinEvol framework is designed to be general. For open-ended generation tasks, we could employ strong LLMs (e.g., GPT-4o) as judges to evaluate the consistency.

**Generation LLM Instruction**, we follow a standardized instruction template as shown in Table 7 to ensure consistent model responses across all benchmarks. This template enforces a structured format for both questions and answers, with explicit markers for answer boundaries to facilitate accurate evaluation. All evaluations utilize the official benchmark scripts to maintain consistency with published results.

**Evaluation Agent Instruction**, we provide the instruction of the Evaluative Agent as shown in Table 8. This provides the Evaluative Agent with a clear directive: given the context of a question and two candidate responses (Response 1 and Response 2), determine if the first is superior to the second and output a simple Y or N. This feedback is the cornerstone that enables our MetaPO algorithm to learn effectively.

## LLMs Usage

We adhere to the ICLR Code of Ethics. We use large language models to polish the text and also fetch the relevant references and the latest related works. The scientific contributions remain entirely our own.

**Case Study: Consistency Check on ConvFinQA**

**Input Prompt:**

Answer the following question. The final answer should be a new line with the following format: 'Answer: ' (without quotes).

*entergy corporation and subsidiaries notes to financial statements as of december 31 , 2008 , system energy had future minimum lease payments ( reflecting an implicit rate of 5.13% ( 5.13 % ) ) , which are recorded as long-term debt as follows : amount ( in thousands ) .*

| Row | Description | Amount |
|---|---|---|
| 2 | 2009 | $ 47760 |
| 3 | 2010 | 48569 |
| 4 | 2011 | 49437 |
| 5 | 2012 | 49959 |
| 6 | 2013 | 50546 |
| 7 | years thereafter | 103890 |
| 8 | total | 350161 |
| 9 | less : amount representing interest | 54857 |
| 10 | present value of net minimum lease payments | $ 295304 |

Question: what portion of the total minimum lease payments is related to interest?
Answer: 0.15666
Question: what about the portion that is due after 2013?

- - - - - - - - - - - - - - - - - - - - - - - - - - - - - - - - - - - - - - - - - - - - - - -

**Model Response 1 (Incorrect):**

To answer the questions, we need to analyze the table.
To calculate the portion of the total minimum lease payments related to interest, we can use the following formula:
Interest / Total = Amount representing interest / Total
Based on the table, the amount representing interest is $54857, and the total is $350161.
Interest / Total = 54857 / 350161 = 0.15666
So, the portion of the total minimum lease payments related to interest is 0.15666.
To calculate the portion that is due after 2013, we can add the amounts for 2014 and onwards (which is the same as the amount for "years thereafter") to the amount for 2013.
Amounts due after 2013 = Amount for 2013 + Amount for years thereafter = 50546 + 103890 = 159336
The total minimum lease payments is $350161.
So, the portion that is due after 2013 is:
159336 / 350161 = 0.45447
**Answer: 0.45447**

**Model Response 2 (Correct):**

To find the portion of the total minimum lease payments that is due after 2013, we first need to identify the total amount due after 2013 and then calculate its proportion of the total minimum lease payments.
From the table provided, the amount due after 2013 is $103,890 (in thousands).
The total minimum lease payments are $350,161 (in thousands).
So, the portion due after 2013 as a fraction of the total is $\frac{103890}{350161}$.
Calculating this gives:

$$\frac{103890}{350161} = 0.2967$$

When expressed as a percentage, this is approximately 29.67%.
Therefore, the portion of the total minimum lease payments that is due after 2013 is 0.2967 or 29.67%.
**Answer: 0.2967**

---

**Consistency Check Process:**
**Rule-based Extraction:** Regex `r"Answer:\s*([0-9.]+)"`

- Response 1 → 0.45447
- Response 2 → 0.2967

Figure 6: Detailed Example of Consistency Check on ConvFinQA.

Table 7: iInstruction template for Generative LLMs.

| Query Template |
|---|
| Answer the following {question_type} question.
Your answer must be on a new line starting with "Answer: ".
{additional_prompt}

Question:
{question} |

Table 8: iInstruction template for the Evaluative Agent ranking task.

| Evaluative Agent Prompt Template |
|---|
| You are a critical ranking agent. Your task is to carefully examine a question and its various predictions to determine the best prediction.

Question: {question}

Response 1: {response_1}
Response 2: {response_2}

Please analyze this carefully and predict if the response 1 is better than response 2.

Your Options:
- Y: Response 1 is better.
- N: Response 2 is better.

Your response must follow this format:

Answer: [Y/N] |

