# OpenReview forum: "Twin Evolution with Meta Preference Optimization for Semi-Supervised Learning of Large Language Models"
_ICLR.cc/2026/Conference — Submitted to ICLR 2026_

### Official Review · Reviewer_eLKh · 2025-10-28

**Soundness:** 3
**Presentation:** 2
**Contribution:** 3
**Rating:** 4
**Confidence:** 4

**Summary:**

This paper presents an approach to semi-supervised training whereby the main LLM is iteratively co-trained with an evaluation agent which provides quality feedback signals for an RL style training. The key idea is two fold: 1) On the labeled examples, model responses are generated and compared with the annotated response. The pairs where the annotation is ranked higher than the model response are used for training the evaluative agent. 2) On the unlabeled examples, K model responses are generated and classified as consistent/inconsistent. SFT is performed on the consistent responses. The evaluative agent generates pseudo-preferences over all pairs of model responses, which are then used for RL using the MetaPO algorithm.  Results are reported using three foundation models: Llama 3.1 8B, Llama 3.2 3B and Gemma 2-9B. on both general purpose and domain benchmarks. The baselines include a number of semi-supervised approaches. The proposed method consistently improves upon the baselines. Ablation studies show that each component in the proposed model plays an important role.

**Strengths:**

* Proposes an approach for semi-supervised learning by iteratively co-training the LLM and an evaluation agent which can provide supervision for RL.
* Shows that the proposed method improves upon several semi-supervised baselines.
* Proposes an extension to the SimPO algorithm (Meng et al. 2024) which uses pseudo-preferences over multiple pairs of responses, and discusses its theoretical properties.

**Weaknesses:**

* The algorithm for identifying consistent vs inconsistent responses is not discussed clearly with examples. This is a crucial component of the proposed algorithm.
* The paper suffers from lack of clarity in multiple places. See questions below.

**Questions:**

* L103: It would be good to compare the proposed method (metaPO) to GRPO (Shao et al. 2024 : https://arxiv.org/pdf/2402.03300)
* L153: What is the algorithm for identifying consistent vs inconsistent responses? It would be useful to give examples of consistent vs inconsistent responses.
* L157: 'We take inconsistent preference annotations as training data for the evaluative agent ' - From the paper, it looks like there is a procedure to classify responses as consistent / inconsistent. However, it is not clear what inconsistent preference annotation means. Could you give an example?
* L179: 'Use the LLM to judge the consistency': Is that done in the paper? If so, what LLM is used for this purpose? Please add details.
* L214: Please discuss how metaPO differs from SimPO. Is SimPO a special case of metaPO?
* L406: "Table 1 shows that SFT and SemiEvol approaches yield substantial improvements by effectively leveraging both labeled and unlabeled data" - Isn't SFT using only labeled data?
* L408: "Post-training techniques show marginal improvements or degradation due to distribution misalign- ment and insufficient capability enhancement." - What are the post-techniques that this sentence is referring to?
* L1052: Please give examples of predefined rules, the consistency validation procedure and examples of consistent vs inconsistent responses.

Typo:
* L061 'while' -> 'with'

---

> ### Author Response · Authors · 2025-11-26
> **Thank you for your thorough review!**
>
> Dear Reviewer eLKh,
>
> Thank you for your thorough review and constructive feedback. We appreciate your valuable suggestions, which have helped us strengthen our paper. We will now address your comments point by point.

---

> ### Author Response · Authors · 2025-11-26
>
> > W1. Clarify consistency check algorithm and add examples in Appendix.
>
> Thank you for your comment. Consistency checking is indeed a crucial component of TwinEvol. We have strengthened the description of this algorithm in Section 3.3 and Appendix D.2. Specifically, we clarified the different checking strategies adopted for different task types (objective questions vs. open-ended generation). For objective questions, we use predefined regex rules to extract answer options or keywords for exact matching. For open-ended generation tasks, we optionally utilize LLMs as judges to evaluate the consistency between the generated content and the reference answer. To illustrate this more intuitively, we have added a specific Case Study in Appendix D.2, showing the complete process of extracting answers from model outputs and comparing them with the Ground Truth.
>
> We have revised Section 3.3 and Appendix D.2.
>
> In Section 3.3:
>
> ```latex
> The consistencies are derived from answer comparisons. For tasks with standard answers (e.g., multiple-choice), we use exact matching based on predefined rules; for open-ended generation, we employ LLM-based judgment to determine consistency. We provide detailed examples of these consistency checks and the corresponding rules in Appendix.
> ```
>
> In Appendix D.2:
>
> ```latex
> Consistency Check Procedure. We generate multiple candidate responses for each unlabeled data point and apply consistency validation. In our main experiments (e.g., MMLU, ARC), we primarily utilize rule-based extraction which is efficient and effective for these structured tasks. However, the TwinEvol framework is designed to be general. For open-ended generation tasks, we could employ strong LLMs (e.g., GPT-4o) as judges to evaluate the consistency.
>
> Case Study: Consistency Check on ConvFinQA
>
> Input Prompt:
> Answer the following question. The final answer should be a new line with the following format: 'Answer: ' (without quotes).
> entergy corporation and subsidiaries notes to financial statements as of december 31 , 2008 , system energy had future minimum lease payments ( reflecting an implicit rate of 5.13\% ( 5.13 \% ) ) , which are recorded as long-term debt as follows : amount ( in thousands ) .
>
> | Row | Description | Amount |
> | --- | --- | --- |
> | 2 | 2009 | \$ 47760 |
> | 3 | 2010 | 48569 |
> | 4 | 2011 | 49437 |
> | 5 | 2012 | 49959 |
> | 6 | 2013 | 50546 |
> | 7 | years thereafter | 103890 |
> | 8 | total | 350161 |
> | 9 | less : amount representing interest | 54857 |
> | 10 | present value of net minimum lease payments | \$ 295304 |
>
> Question: what portion of the total minimum lease payments is related to interest?
> Answer: 0.15666
> Question: what about the portion that is due after 2013?
>
> ---
>
> Model Response 1 (Incorrect):
> To answer the questions, we need to analyze the table.
>
> To calculate the portion of the total minimum lease payments related to interest, we can use the following formula:
>
>  Interest / Total = Amount representing interest / Total
>
> Based on the table, the amount representing interest is \$54857, and the total is \$350161.
>
>  Interest / Total = 54857 / 350161 = 0.15666
>
> So, the portion of the total minimum lease payments related to interest is 0.15666.
>
> To calculate the portion that is due after 2013, we can add the amounts for 2014 and onwards (which is the same as the amount for "years thereafter") to the amount for 2013.
>
> Amounts due after 2013 = Amount for 2013 + Amount for years thereafter
> = 50546 + 103890
> = 159336
>
> The total minimum lease payments is \$350161.
>
> So, the portion that is due after 2013 is:
>
> 159336 / 350161 = 0.45447
>
> Answer: 0.45447
>
> ---
>
> Model Response 2 (Correct):
> To find the portion of the total minimum lease payments that is due after 2013, we first need to identify the total amount due after 2013 and then calculate its proportion of the total minimum lease payments.
>
> From the table provided, the amount due after 2013 is \$103,890 (in thousands).
>
> The total minimum lease payments are \$350,161 (in thousands).
>
> So, the portion due after 2013 as a fraction of the total is $\frac{103890}{350161}$.
>
> Calculating this gives:
>
> $$\frac{103890}{350161} = 0.2967$$
>
> When expressed as a percentage, this is approximately $29.67\%$.
>
> Therefore, the portion of the total minimum lease payments that is due after 2013 is $0.2967$ or $29.67\%$.
>
> Answer: 0.2967
>
> ---
>
> Consistency Check Process:
>
> Rule-based Extraction: Regex r"Answer:\s*([0-9.]+)"
>
> Response 1 $\rightarrow$ 0.45447
> Response 2 $\rightarrow$ 0.2967
>
> Inconsistent.
> ```

---

> ### Author Response · Authors · 2025-11-26
>
> > Q1. Discuss GRPO in Related Work.
>
> Thank you for the suggestion. GRPO is a relevant work. MetaPO differs from it:
>
> - **Different Task Scenarios:** GRPO is mainly used in the RL stage, focusing on improving reasoning capabilities through Verifier feedback; while MetaPO is designed for semi-supervised fine-tuning, aiming to utilize limited labeled data and abundant unlabeled data for domain adaptation.
> - **Different Reward Mechanisms:** GRPO relies on Verifiers and uses group-wise comparison to estimate advantages; MetaPO utilizes a co-evolved Evaluative Agent to provide pair-wise preference ranking, which is more general.
> - **Different Optimization Objectives:** MetaPO focuses on mining hard negative sample pairs from multiple generated responses and optimizing through a Pairwise objective function to leverage subtle differences in unlabeled data.
>
> We have added a discussion and comparison of GRPO in the Related Work section.
>
> We have added the discussion in Preliminaries:
>
> ```latex
> GRPO leverages group-wise comparisons but focuses on reinforcement learning to enhance long-chain reasoning. Unlike GRPO which relies on rule-based verifiers and optimizes a group-relative objective, MetaPO is designed for semi-supervised adaptation, utilizing a co-evolved Evaluative Agent to provide pairwise preference signals from multiple generated responses.
> ```
>
> > Q2. Explain consistency check algorithm at L153.
>
> Thank you for your question. Regarding the consistency identification algorithm, our core logic is to compare the model-generated answer $\hat{Y}$ with the standard answer $Y$. Specifically:
>
> - For objective tasks, we use predefined regex rules to extract answer options or keywords for exact matching. For example, for ConvFinQA, we extract "0.2967" from "Answer: 0.2967" and compare it with the standard answer (within error tolerance).
> - For open-ended generation tasks, we optionally utilize LLMs as judges to evaluate the consistency between the generated content and the reference answer.
>
> We have clarified this process in Section 3.3 and provided specific examples in Appendix D.2.
>
> > Q3. Clarify definition of inconsistent preference annotations.
>
> Thank you for your question. "Inconsistent preference annotations" refers to cases where the model's generated response $\hat{Y}$ is inconsistent with the Ground Truth $Y$ (usually implying lower quality than the Ground Truth). In such cases, we utilize this inconsistency to construct training signals:
>
> - We treat the Ground Truth $Y$ as the positive sample (Winner).
> - We treat the model's erroneous response $\hat{Y}$ as the negative sample (Loser).
> - We thus construct a preference pair $(Y \succ \hat{Y})$ to train the Evaluative Agent, enabling it to learn to distinguish between good and bad responses.
>
> We have revised Section 3.2 to make the definition clearer.
>
> > Q4. Specify the LLM used for consistency check in Appendix.
>
> Thank you for your question. In our main experiments (MMLU, MMLU-Pro, ARC, FPB, USMLE, PubMedQA, ConvFinQA), since these tasks mostly have clear answer standards, we primarily adopted rule-based extraction for consistency checking, which is sufficiently effective. However, it is worth emphasizing that the TwinEvol framework is designed to be general, and different Judge methods (rules or LLMs) do not affect the framework's ability to function through the co-evolution mechanism. We have supplemented specific model details in the Appendix.
>
> > Q5. Clarify relation between MetaPO and SimPO.
>
> Thank you for your question. Yes, SimPO can be viewed as a special case of MetaPO where $K=2$ (i.e., using only one pair of preference data). MetaPO generalizes this to the case of $K>2$, which is particularly important in semi-supervised scenarios:
>
> - **Rich Negative Samples:** MetaPO utilizes $K>2$ generated responses to construct multiple negative sample pairs, mining finer-grained preference differences.
> - **Stability and Variance Reduction:** As proved in Theorem 3.1, by aggregating signals from multiple preference pairs, MetaPO significantly reduces gradient variance. This is crucial for stabilizing model training in noisy semi-supervised environments.
>
> We have clarified this in Section 3.4.2.

---

> ### Author Response · Authors · 2025-11-26
>
> > Q6. Clarify SFT uses labeled data only in Section 5.2.
>
> Thank you for your question. Yes, SFT uses only labeled data. Our original intention was to emphasize that SFT (using labeled data) establishes a foundation, while SemiEvol/TwinEvol achieves greater improvements by further utilizing unlabeled data. We have modified the sentence to eliminate ambiguity and clearly distinguish the data usage of SFT and Semi-supervised methods.
>
> We have clarified this in Section 4.2.
>
> > Q7. Explicitly name the post-training techniques in Section 5.2.
>
> Thank you for your question. The "Post-training techniques" here specifically refer to two categories of methods listed in Table 1:
> (1) Self-Evolution methods: such as Reflection-Llama, Hermes-3. (2) Domain Adaptation methods: such as AdaptLLM.
>
> These methods showed limited improvement in our experimental settings (semi-supervised, domain adaptation), mainly because they struggle to adapt to distribution shifts without targeted supervision signals. We have explicitly listed the names of these methods to avoid confusion.
>
> > Q8. Provide examples of predefined rules and consistency validation.
>
> Thank you for your question. To address your request, we have supplemented detailed examples of predefined rules in Appendix D.2.
>
> - We use the regular expression to extract the answer from the model output.
> - We have displayed a complete Consistency Validation flow chart, intuitively showing how to determine whether a response is Consistent or Inconsistent.
>
> > Typo. Fix typo at L061.
>
> Thank you for the correction. We have fixed this typo.

---

> ### Author Response · Authors · 2025-11-26
>
> We are grateful for your constructive feedback. We have carefully revised the manuscript to address your concerns on consistency check algorithms, related work comparisons, and experimental clarifications. We hope these revisions are to your satisfaction!

---

### Official Review · Reviewer_Bmhn · 2025-10-29

**Soundness:** 3
**Presentation:** 3
**Contribution:** 3
**Rating:** 4
**Confidence:** 4

**Summary:**

This paper introduces TwinEvol, a framework for semi-supervised learning of LLMs designed to address the challenge of limited labeled data. The core of TwinEvol is a dual-branch architecture where a generative LLM and an "Evaluative Agent" co-evolve. This process creates a self-reinforcing cycle that transfers knowledge from a small amount of labeled data to a larger pool of unlabeled data. Key components of this framework include Hard Negative Mining and a MetaPO algorithm. The authors provide theoretical analyses of their approach and empirical results across several benchmarks, demonstrating that TwinEvol outperforms existing methods.

**Strengths:**

1. The concept of a co-evolving generative LLM and Evaluative Agent is a creative and promising approach to semi-supervised learning. This "twin evolution" creates a dynamic and self-improving system which embodies the idea of self-play and DPO, expected to be able to continuously enhance its capabilities.

2. The paper presents a comprehensive set of experiments on various benchmarks, including MMLU, MMLU-Pro, ARC, FPB, USMLE, PubMedQA, and ConvFinQA. The results consistently show that TwinEvol outperforms a range of baseline methods, including vanilla SFT, self-evolution approaches, and domain adaptation techniques. The improvement is also consistent across different model architectures and scales (Llama3.1-8B, Gemma2-9B, and Llama3.2-3B).

**Weaknesses:**

1. Complexity of the Framework: The TwinEvol framework, with its dual-branch architecture, co-evolutionary loop, and multiple components (Hard Negative Mining, MetaPO), is considerably more complex than standard fine-tuning methods. The practical overhead of implementing and tuning such a system could be a barrier to adoption.

2. The most critical concern is if the proposed "twin evolution" mechanism is really effective. On all benchmarks, TwinEvol outperforms baseline SemiEvol, and even the naive SFT, only by very small margin. Considering the numerous techniques implemented above, and results in ablation study (Table 3), the reviewer question that the "twin evolution" mechanism itself brings no improvement. The marginal performance gain is due to some technique implementations only. For example, it seems that consistency checking is very effective itself comparing V1 and V2. The reviewer wonder if the performance of TwinEvol would degrade a. lot without CONS, or  other baselines can be improved simply with CONS.

**Questions:**

Please refer to weakness.

---

> ### Author Response · Authors · 2025-11-26
> **Thank you for your thorough review!**
>
> Dear Reviewer Bmhn,
>
> Thank you for your thorough review and constructive feedback. We appreciate your valuable suggestions, which have helped us strengthen our paper. We will now address your comments point by point.

---

> ### Author Response · Authors · 2025-11-26
>
> > W1. Address framework complexity concern
>
> Thanks for your comment. We will discuss from the following points:
>
> - **Training vs. Inference.** The complexity exists only during training. In inference, TwinEvol deploys only the Generative Model (no Evaluative Agent), maintaining identical efficiency to standard SFT with no additional cost.
>
> - **Design Rationale.** TwinEvol automates supervision mining from unlabeled data, trading training-time computation for substantial annotation cost reduction—a favorable trade-off in data-scarce scenarios.
>
> - **Empirical Overhead.** Appendix C.1 shows training time increases marginally vs. SFT, much lower than MemoryLLM (3.02h). Code is open-sourced with modular design.
>
> ---
>
> > W2. Validate twin evolution mechanism's independent contribution
>
> Thank you for your question. We use our ablation study to demonstrate the independent contribution of Twin Evolution beyond CONS.
>
> - **CONS as an Effective Component.** We acknowledge that CONS is an effective component within TwinEvol. Table 3 shows V2 (SFT+CONS) outperforms V1 (SFT alone), confirming CONS's value in filtering low-quality pseudo-labels.
>
> - **Critical Evidence: The V5 Experiment.** To isolate Twin Evolution's contribution, we designed V5 as a controlled experiment. V5 uses a **static evaluator** with MPO, CONS, and SFT. If gains came primarily from CONS, V5 should **match or exceed V2**. However:
>
> | Variant  | Configuration           | MMLU                  | MMLU-Pro | ARC  |
> | -------- | ----------------------- | --------------------- | -------- | ---- |
> | **V2**   | SFT + CONS              | 68.9                  | 50.2     | 83.3 |
> | **V5**   | V2 + MPO (Static Eval)  | 67.1 (**-1.8**)       | 49.8     | 82.5 |
> | **Full** | V2 + MPO (Co-evolution) | 69.7 (**+2.6** vs V5) | 50.8     | 84.7 |
>
> - **Why V5 Fails.** This counterintuitive result reveals a fundamental problem: static evaluators cause **alignment drift** in semi-supervised learning. As the Generative Model evolves, the fixed evaluator's preference signals become increasingly misaligned, making preference optimization harmful rather than helpful.
>
> - **Why TwinEvol(Full) Succeeds.** TwinEvol's co-evolution mechanism solves this problem. By dynamically adapting the evaluator alongside the generator, Full successfully reverses V5's negative effect, achieving a large gain over V5 (67.1→69.7) and consistent improvements over V2 across all benchmarks. This demonstrates that Twin Evolution is not merely incremental. It transforms harmful static preference learning into beneficial dynamic optimization.
>
> To summarize, CONS and Twin Evolution address different challenges:
>
> - **CONS**: Denoising at the data level (filtering bad positives)
> - **Twin Evolution**: Mining at the learning level (dynamic preference optimization)
>
> Both components are essential. Without CONS, data noise overwhelms any learning method. Without Co-evolution, preference optimization degrades performance due to distribution shift.
>
> We have updated the Methodology accordingly.

---

> ### Author Response · Authors · 2025-11-26
>
> We are grateful for your constructive feedback. We have carefully revised the manuscript to address your concerns on framework complexity and the independent contribution of the twin evolution mechanism. We hope these revisions are to your satisfaction!

---

### Official Review · Reviewer_XipS · 2025-11-03

**Soundness:** 3
**Presentation:** 3
**Contribution:** 2
**Rating:** 4
**Confidence:** 2

**Summary:**

The paper proposes a training framework in which the LLM and the evaluator are trained alternatingly. The goal is semi-supervised training for leveraging both unlabelled and labelled data. The EvaluationAgent is trained on labelled examples on which it disagrees with the reference. The generative LLM is trained on labelled data first, and then samples output on unlabelled data and uses the signal from the EvaluationAgent.

**Strengths:**

- The high-level setup makes intuitive sense
- Quite consistent gains across multiple models and datasets (but see caveat below)

**Weaknesses:**

I feel that the substance of this work is harder to extract than it needs to due to a quite extensive use of terms like knowledge/evolutionary flywheel, co-evolutionary loop/framework, information infusion/propagation/injection in contexts where it's not helpful - a more concise and sober description would help clarity.

At the same time, there are a couple of points that stay fairly vague. For example, the way D_incon and D_con are split is not described sufficiently (L187), although it seems to be a crucial part of the approach.

IIUC the baselines in Table 1 are of similar size, but they are not based on exactly the same LLM. This would mean that the only apples-to-apples comparison is against vanilla and SFT - adding more sophisticated baselines to Table 2 would strengthen the results.

**Questions:**

- Can you add stronger baselines to Table 2?

---

> ### Author Response · Authors · 2025-11-26
> **Thank you for your thorough review!**
>
> Dear Reviewer XipS,
>
> Thank you for your thorough review and constructive feedback. We appreciate your valuable suggestions, which have helped us strengthen our paper. We will now address your comments point by point.

---

> ### Author Response · Authors · 2025-11-26
>
> > W1. Please simplify the terminology for clarity.
>
> Thank you for your suggestion. We agree that clear and concise terminology is essential for effectively communicating our core contributions. We have thoroughly reviewed the manuscript, particularly the Abstract and Introduction, and simplified metaphorical terms like _knowledge flywheel_ and _information infusion_. In the revised version, we employ more direct and objective descriptions to explain our framework. These changes make the underlying mechanism of our methodology more prominent and easier to understand.
>
> We have revised the manuscript in the Abstract and Section 1 accordingly.
>
> > W2. Please clarify the splitting mechanism of consistent and inconsistent sets.
>
> Thank you for your comment. Accurately distinguishing between consistent and inconsistent sets is indeed a critical step in our method. We have added detailed implementation specifics in Section 3.3 of the revised manuscript. Specifically, we adopt a task-dependent strategy for consistency checking: for tasks with standard answers (e.g., multiple-choice or numerical questions), we use exact matching or rule-based parsing; for open-ended generation tasks, we utilize semantic similarity checks or LLM-based consistency judgment. This ensures that the splitting process is both rigorous and adaptable to different task scenarios.
>
> We have revised the manuscript in Section 3.3 accordingly.
>
> > W3. Please add stronger baselines for a fairer comparison.
>
> Thank you for your comment. We understand your concern about fair comparison. First, we would like to clarify that the core comparisons in our main results (Table 2), were all re-implemented using the **exact same foundation models** (Llama-3.1-8B) to ensure a strict _apples-to-apples_ comparison. Other baselines like AdaptLLM are included as reference points for models of similar scale.
>
> To further strengthen our results, we have added two strong baselines, `Reflection` (using reflection-llama dataset to SFT) and `SemiEvol`, to **Table 3** for all foundation models (Llama-3.2-3B, Llama-3.1-8B, and Gemma-2-9B). The results show that while SemiEvol provides improvements over SFT, TwinEvol consistently outperforms it due to our co-evolutionary mechanism.
>
> We have revised the manuscript in Table 3 and Section 4.3 accordingly.
>
> | Method        |   MMLU   | MMLU Pro |   ARC    |   FPB    |  USMLE   | PubMedQA | ConvFinQA |   Avg.   |
> | :------------ | :------: | :------: | :------: | :------: | :------: | :------: | :-------: | :------: |
> | _Llama3.2 3B_ |          |          |          |          |          |          |           |          |
> | Vanilla       |   59.2   |   22.4   |   68.1   |   62.0   |   40.1   |   59.0   |   28.8    |   48.5   |
> | SFT           |   61.2   |   41.1   |   73.9   |   92.8   |   63.9   |   71.0   |   65.8    |   67.1   |
> | Reflection    |   60.1   |   30.2   |   72.4   |   62.0   |   42.9   |   58.5   |   23.2    |   49.9   |
> | SemiEvol      |   62.0   |   41.6   |   75.1   |   93.4   |   64.2   |   72.5   |   68.0    |   68.1   |
> | **TwinEvol**  | **63.1** | **42.3** | **76.5** | **94.1** | **65.8** | **74.0** | **70.2**  | **69.4** |
> | _Llama3.1 8B_ |          |          |          |          |          |          |           |          |
> | Vanilla       |   66.4   |   47.1   |   81.1   |   81.7   |   70.2   |   73.5   |   51.1    |   67.3   |
> | SFT           |   67.9   |   49.8   |   81.8   |   96.2   |   70.8   |   75.0   |   81.3    |   74.7   |
> | Reflection    |   65.5   |   37.5   |   82.2   |   80.8   |   67.4   |   71.5   |   40.8    |   63.7   |
> | SemiEvol      |   68.8   |   50.3   |   83.4   |   96.2   |   71.6   |   76.0   |   82.4    |   75.5   |
> | **TwinEvol**  | **69.7** | **50.8** | **84.7** | **97.5** | **72.0** | **77.0** | **84.1**  | **76.5** |
> | _Gemma2 9B_   |          |          |          |          |          |          |           |          |
> | Vanilla       |   72.1   |   43.0   |   87.1   |   73.7   |   58.7   |   64.5   |   45.9    |   63.6   |
> | SFT           |   73.7   |   50.0   |   87.6   |   95.2   |   66.6   |   74.5   |   79.8    |   75.3   |
> | Reflection    |   73.0   |   41.5   |   86.5   |   75.5   |   58.7   |   66.2   |   50.1    |   64.5   |
> | SemiEvol      |   74.5   |   50.8   |   88.0   |   96.0   |   67.3   |   75.3   |   81.5    |   76.2   |
> | **TwinEvol**  | **75.4** | **51.9** | **88.6** | **96.5** | **68.4** | **76.8** | **83.7**  | **77.3** |

---

> ### Author Response · Authors · 2025-11-26
>
> We are grateful for your constructive feedback. We have carefully revised the manuscript to address your concerns on terminology clarity, consistency check implementation, and baseline comparisons. We hope these revisions are to your satisfaction!

---

### Official Review · Reviewer_cXH8 · 2025-11-10

**Soundness:** 3
**Presentation:** 2
**Contribution:** 3
**Rating:** 6
**Confidence:** 3

**Summary:**

The paper presents and effort to develop a joint evolution framework, termed as TwinEvol, where an evaluation agent and the base LLM are jointly optimized. The authors use a mixture of supervised and semi-supervised data to perform the joint optimization.

The work provides certain theoretical guarantees, although they largely reflect SimPO and other DPO guarantees.

On the evaluation front, the authors provide a comparison with SFT and other methods and highlight some of the improvements achieved.

In summary, paper does point to interesting methods to jointly develop an evaluation agent along with the LLM fine-tuning. However, the evaluation and the impact are somewhat unclear.

**Strengths:**

Proposing the LLM fine-tuning as a joint optimization framework which also involves the learning of an evaluation agent.

Establishing a theoretical framework to analyze the MetaPO in conjunction with other works based on preference optimization.

Evaluation on multiple tasks and models.

**Weaknesses:**

The major issue with the paper is the establishment of the impact and the unfair comparison that the authors have made in the evaluations.

  i. As part of the TwinEvol, the labeled data is used for regular SFT. However, the proposed framework also uses unlabeled data. The amount of unlabeled data used in their setting is not specified either, while this setup is refereed to a prior work. In Table 2, the performance comparison is provided only for the SFT (which uses supervised data and is also part of the proposal). As the TwinEvol also requires additional unsupervised data, this comparison with SFT is unfair.

Ii. A related concern is in the strategy proposed to utilize the unsupervised data. In Section 3.3, the authors mention the division of the unsupervised pile into a consistent set and an inconsistent set. The authors mention, rather vaguely, the use of “answer comparisons”, “LLM to judge the consistency”. What LLM was used for this purpose and how. The augmentation of this additional “LLM as judge” data could also improve the SFT and other semi-supervised strategies. What additional value does MetaPO bring. The description of using unsupervised data is rather shallow and leaves many questions unanswered.

Iii. The prior work citations and comparisons do not dwell deeply into various related efforts in the field of semi-supervised fine-tuning. For example, efforts related to self-alignment with LLM as a judge [“Meta Rewarding Language Models …” by Wu et al. 2024], Semi-supervised reward modeling [He et al 2024], or self-rewarding language models [SRLM, ICML 2024] are already available. The authors do not compare with these prior efforts, nor is there reference to other similar works on semi-supervised modeling in preference optimization. Without these comparisons and benchmarking, the impact and value addition with the proposal is unclear. Further, the positioning of the work with respect to prior art is also well established.

Iv. The computational complexity discussion leaves a lot to be desired. Does the comparison with SFT and proposal (Table 4) also include the cost of the SFT for the TwinEvol, as the proposal requires SFT as the starting point. Further, how does this compare with other approaches like SemiEvol which utilize unsupervised data.

v. The writing has various typos and errors, which warrants a thorough proof-reading. Further, various notations are introduced without fully describing the components. In this regard, it appears that the paper writing was performed in haste, leaving it somewhat unpolished.

**Questions:**

— Careful benchmarking with other prior efforts that utilize semi-supervised learning frameworks, where LLM-as-a-judge is employed.
— Clear description of the efforts in utilizing the semi-supervised data and how the LLM-as-a-judge is utilized in the proposal.
— Appropriate computational considerations which bring out the exact requirements of the proposal.
— Positioning the work in a refined manner to understand the value addition of the work.
— Considerations in creating the Evaluation Agent and ablations around the errors made by the evaluation agent itself. For example, the paper does not dwell on how accurate the evaluation agent alone is, and what impact this has on the rest of the LLM training pipeline.
-- Thorough rewriting to improve the descriptive quality of the draft.

---

> ### Author Response · Authors · 2025-11-26
> **Thank you for your review!**
>
> Dear Reviewer cXH8,
>
> Thank you for your thorough review and constructive feedback. We appreciate your valuable suggestions, which have helped us strengthen our paper. We will now address your comments point by point.

---

> ### Author Response · Authors · 2025-11-26
>
> > W1&Q1. Address the unfair comparison with SFT and specify unlabeled data usage.
>
> We would like to clarify two key points regarding fairness:
>
> 1.  **Baseline Selection.** While SFT uses only labeled data, we include it as a lower bound to demonstrate the gain from unlabeled data. Crucially, we compare against **SemiEvol** (Table 2), a SOTA semi-supervised method. SemiEvol and TwinEvol use **identical** labeled and unlabeled data settings. TwinEvol significantly outperforms SemiEvol, proving that gains stem from our Co-evolution mechanism, not just data quantity.
> 2.  **Data Usage.** We use the **complete** unlabeled training set for all semi-supervised methods (TwinEvol, SemiEvol). We have explicitly stated this in Section 5.1 and Appendix D.2.
>
> We have updated the paper to make these points clearer.
>
> > W2&Q2. Clarify the consistency check details (referencing the new Case Study), the LLM judge used, and the value of MetaPO.
>
> Thank you for your question. We employ task-specific strategies for consistency checks: rule-based Regex matching for objective tasks due to its efficiency, and LLM-as-a-judge for open-ended generation to evaluate consistency with the ground truth. To visualize this process, we have added a detailed Case Study in Appendix.
>
> Regarding the value of MetaPO, while LLM-as-a-judge can indeed filter data for SFT (similar to Rejection Sampling), MetaPO offers unique advantages. Unlike SFT which only learns from positive samples, MetaPO leverages pairwise comparisons to teach the model "why A is better than B," utilizing rich signals from negative samples. Furthermore, our theoretical analysis proves that MetaPO significantly reduces gradient variance compared to standard methods, which is critical for stabilizing training with noisy semi-supervised signals.
>
> We have revised the paper accordingly.
>
> > W3&Q4. Compare with related works on semi-supervised learning and self-alignment (Self-Rewarding, Meta-Rewarding, Semi-supervised RM).
>
> Thanks for your comment. We have added discussions on [1,2,3] in the Related Work section.
>
> **Key Distinctions:**
>
> - **Task & Data:** [2] and [3] focus on general instruction following via self-generated prompts. TwinEvol targets semi-supervised downstream adaptation, solving data scarcity in specific domains using fixed unlabeled data.
> - **Mechanism:** Unlike [3] (single model) or [1] (iterative self-training), TwinEvol features Twin Co-evolution. We decouple Generator and Evaluator, allowing them to serve as a mutual curriculum.
> - **Optimization:** We propose MetaPO specifically for noisy pseudo-pairs. Unlike standard DPO used in [2][3] or thresholding in [1], MetaPO aggregates multi-sample preferences to reduce variance (Theorem 1), offering superior robustness.
>
> [1] Semi-supervised reward modeling via iterative self-training. EMNLP Find 2024.
>
> [2] Meta-rewarding language models: Self-improving alignment with llm-as-a-meta-judge. EMNLP 2025.
>
> [3] Self-rewarding language models. ICML 2024.
>
> We have modified the related work section to include these works.
>
> > W4. Clarify computational complexity including SFT costs and compare with semi-supervised baselines.
>
> Thanks for your comment. We have clarified in Appendix C.1 that our reported time includes all training overhead (including the Evaluative Agent). Initial SFT is a one-time prerequisite shared by all fine-tuning methods (including baselines). TwinEvol incurs only a marginal computation increase due to dual-branch updates but yields significant performance gains.
>
> We have updated the paper to make this point clearer.
>
> > W5. Proofread the paper to fix typos and clarify notations.
>
> We have thoroughly proofread the paper, correcting typos and unifying mathematical notations and definitions to enhance clarity.
>
> > Q5. Analyze the Evaluation Agent's accuracy and the impact of its errors on training.
>
> Thanks for the comment. We will dicuss fromt the following points:
>
> 1.  **Dynamic Accuracy.** The Agent's accuracy is not static; it improves via calibration on labeled data and co-evolution with the main model.
> 2.  **Robustness Analysis.** We added a **Robustness Analysis** in Appendix C.3, simulating Evaluator errors (noise). Results (Table below) show TwinEvol maintains superior performance even with **30% noise**, proving MetaPO's ability to mitigate individual errors through aggregated preference signals.
>
> **Table: Robustness to semantic noise on ConvFinQA (Accuracy, %)**
>
> | Method        |    0%    |   10%    |   20%    |   30%    |
> | :------------ | :------: | :------: | :------: | :------: |
> | SFT           |   81.3   |    -     |    -     |    -     |
> | SFT+SelfLabel |   81.9   |   80.9   |   79.5   |   77.8   |
> | SemiEvol      |   82.4   |   82.0   |   81.3   |   79.6   |
> | **TwinEvol**  | **84.1** | **83.6** | **83.1** | **82.9** |
>
> We have modified the paper accordingly.

---

> ### Author Response · Authors · 2025-11-26
>
> We are grateful for your constructive feedback. We have carefully revised the manuscript to address your concerns on fairness of comparisons, consistency check details, and related work. We hope these revisions are to your satisfaction!

---

### Author Response · Authors · 2025-11-26
**Revision Summary**

We sincerely thank all reviewers for their insightful feedback. We have carefully addressed all concerns and made substantial revisions to improve the manuscript.

- **Clarified Terminology**: Simplified metaphorical terms in the Abstract and Introduction, adopting more direct and objective descriptions to explain our framework.

- **Enhanced Consistency Check Details**: Detailed the task-specific consistency check strategies (rule-based vs. LLM-based) in the Methodology and Appendix, and added a concrete Case Study to visualize the process.

- **Enhanced Related Work**: Added discussions on Self-Rewarding LMs, Meta-Rewarding LMs, Semi-supervised Reward Modeling, and GRPO, clarifying TwinEvol's unique position in semi-supervised downstream adaptation.

- **Strengthened Experimental Validation**: Added the Reflection baseline to the main results and clarified the fair comparison with the SOTA semi-supervised method SemiEvol (using identical data), highlighting the superiority of the co-evolution mechanism.

- **Added Robustness & Efficiency Analysis**: Included a robustness analysis against evaluator noise to demonstrate MetaPO's stability, and clarified computational overhead to prove the method's efficiency.

We are grateful for the opportunity to improve our paper with this feedback. All revisions are marked in the manuscript. For our point-by-point responses to each reviewer, please see the respective threads.

---

### Meta-Review · Area_Chair_zJ14 · 2026-01-02

**Summary:**

Here are the main concerns of the reviewers:
* the writing is missing details and procedures of the approach are not well explained; lack of clarity
* questioning whether the method is scalable or too complex
* questioning whether the numerical experiments are meaningful in terms of similar baselines.

**Reviewer Concerns:**

I believe the authors did address the concerns.

**Reviewer Scores:**

It’s really hard to say how any reviewer would have changed their score if they had taken part more fully in the discussion. Without hearing it from them directly, anything we write here would just be guesswork.

For this paper, the scores were 6,4,4,4. However, the strengths mentioned by the reviewers are to my mind not sufficient for warranting acceptance, even from Reviewer cXH8, who gave the score 6. Hence I do not believe that the reviewers would have changed their scores significantly, maybe only by one, which would leave it in the rejection area.

---

### Decision · Program_Chairs · 2026-01-26

Reject